# Three Years of r/ChatGPT:
# Societal Impact Evaluations from Social Media Data

**Jessica Dai** [1]  **Sean Garcia** [1]  **Emma Pierson** [1]  **Benjamin Recht** [1]  **Nika Haghtalab** [1]

## Abstract

ChatGPT was launched on November 30, 2022; the r/ChatGPT subreddit was created just one day later. Since then, chatbot-based AI products have gone from niche proofs-of-concept to widely-used household names. However, the ways in which adoption has developed among the public remains poorly understood. In this paper, we develop a framework for using social media as a data source for understanding the societal impact of widely-adopted consumer AI products, and propose PULSE (*Public and Longitudinal Signals for Evaluation*), a general approach to monitoring for societally-impactful trends in real time. We apply our framework to conduct what is, to the best of our knowledge, the first longitudinal study of r/ChatGPT. We find that, overall, r/ChatGPT posts over time illustrate the normalization of ChatGPT as an everyday consumer product rather than an exceptional, novel technology. However, our retrospective analysis also finds that posts about using ChatGPT for mental health support, and posts about developing emotional attachments to ChatGPT, both rise steadily in frequency almost immediately after the launch of GPT-4o in May 2024. We show that PULSE can detect the increase in emotional engagement as early as October 2024—months before OpenAI made any (public) acknowledgment of this impact.

*An interactive site to explore our results and methods, updated daily with live data, is available at* **rchatgpt-pulse.github.io**.

[1]University of California, Berkeley. Correspondence to: Jessica Dai <jessicadai@berkeley.edu>.

*Proceedings of the $43^{rd}$ International Conference on Machine Learning*, Seoul, South Korea. PMLR 306, 2026. Copyright 2026 by the author(s).

[1]openai.com/index/sycophancy-in-gpt-4o/

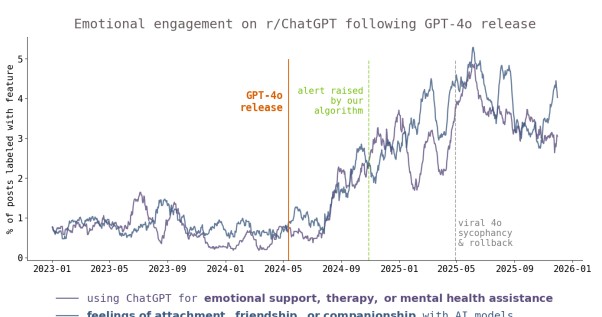

Emotional engagement on r/ChatGPT following GPT-4o release

*Figure 1.* Posts about both AI therapy and AI companionship begin to rise in frequency almost immediately after the release of GPT-4o. We propose a real-time monitoring method (Section 4) that could have detected this as early as October 2024; in contrast, GPT-4o's behavior did not reach the level of public discourse until April 2025, when an extremely-sycophantic update triggered a rollback.[1]

## 1. Introduction

The launch of ChatGPT in late 2022 was a watershed moment for consumer AI products: ChatGPT reflected a step-change not only in the capabilities of AI products available to the general public, but in the degree to which any LLM-based product reached widespread consumer adoption. Now, a little more than three years after ChatGPT's launch, this recent history can be studied with the benefit of hindsight. To this end, recent works have sought to understand the realized impact of deploying LLM-based products on domains such as education, labor, and healthcare (e.g., Bastani et al., 2025; Brynjolfsson et al., 2025; Goh et al., 2024).

Domain-specific evaluations naturally give rise to well-defined measurement targets that can be pre-specified and tracked over time. However, a technology with a user base approaching one billion will inevitably have unpredictable effects. How might we identify—and study—such effects?

In this work, we turn to social media: beyond adoption, ChatGPT is also unique in the extent to which its rollout has been "online." Its users are highly active on social media—in fact, its early and explosive success can be attributed at least in part to virality on platforms like Twitter/X and Reddit. This makes social media a natural source of data for studying the societal impacts of ChatGPT in particular.

Our approach relies on the core assumption that social media posts from everyday users of a technology reflect those users' perceptions and priorities about that technology—that is, that social media provides signal about "societal impact." However, what those perspectives actually entail is unknown *a priori*. Our framework thus begins with an unsupervised step to identify potentially-relevant ideas surfaced among all posts. Our key proposal to formalize *impact* is to explicitly track how these concepts develop over time; this can be quantified by placing temporal behavior in context with known external events, such as model and product releases.

To the best of our knowledge, ours is the first longitudinal analysis of r/ChatGPT over this time period. Our substantive findings in Section 3 tell two parallel stories of adoption. On the one hand, ChatGPT has become normalized as a tool that is a part of users' routine workflows for everyday tasks. On the other, emotional engagement with ChatGPT also emerges as an increasingly compelling use-case; this appears to be driven in large part by the GPT-4o model, which was released in May 2024.

One natural question is whether we might have known about these impacts sooner—and if so, how. Therefore, in Section 4 we also provide a *pro*spective approach to real-time monitoring, which we call PULSE (*Public and Longitudinal Signals for Evaluation*). PULSE discovers statistically meaningful growth in emotional engagement as early as October 2024—long before OpenAI took any public action regarding the emotional-health impacts of their product (see Appendix A.2 for discussion of what was "known," and by whom, at various points in time).

## 1.1. Related work

Our work makes use of the rich methodologies that have been developed for clustering, topic modeling, and other unsupervised approaches (e.g., Blei & Lafferty, 2006, or more recently, Pham et al., 2024; Reuter et al., 2024). We use sparse autoencoders (SAEs), which have recently emerged as compelling methods for analyzing text-as-data (see, e.g., Jiang et al., 2025; Movva et al., 2025; Peng et al., 2025); however, our methods are agnostic to what specific algorithm is used as long as the outputs of the method are consistent with what is outlined in Section 2.

Modeling the dynamics of online content over time is also a canonical problem (e.g., Leskovec et al., 2009; Danescu-Niculescu-Mizil et al., 2013); more recently, Desiderio et al. (2025) study the dynamics of Reddit conversations in response to external events. Event and topic detection from social media is also well-studied (see, e.g., surveys in Karimiziarani, 2022; Atefeh & Khreich, 2015; Asgari-Chenaghlu et al., 2021). Typical methods involve heuristic approaches to sequential clustering (e.g., Kolajo et al., 2022; McCreadie et al., 2013; Li et al., 2017; Aiello et al., 2013; Fedoryszak

et al., 2019; Qiu et al., 2025) and algorithms that identify "bursts" in specific topics or keywords (e.g., Mathioudakis & Koudas, 2010; Xie et al., 2016; Shamma et al., 2011). A subtle challenge that distinguishes our setting is that while prior work typically focuses on correctly identifying "trending" or "bursty" topics only in the moment, we are interested in tracking long-run changes over time, not just short-term effects.

The substantive findings we present in Section 3 build on prior works about social media posts about ChatGPT, including Twitter (Demirel et al., 2025) and Reddit, that have used both quantitative (e.g., Xu et al., 2024; Qutieshat, 2024) and qualitative (e.g., Choi et al., 2023) approaches. Jung et al. (2025) explicitly studies posts about mental health on r/ChatGPT. Prior work has also analyzed subreddits more specific to emotional relationships, such as r/ReplikaOfficial and r/MyBoyfriendIsAI (e.g., Hanson & Bolthouse, 2024; Depounti et al., 2023; Tunca, 2025; Pataranutaporn et al., 2025); our findings are complementary to (and consistent with) these. To the best of our knowledge, our work is the first to study r/ChatGPT with three years of data, and with the question of temporal variation explicitly in mind.

A primary goal for this work is to serve as a longitudinal evaluation of the ChatGPT product. Prior works (e.g., Chen et al., 2024; Cen et al., 2025) have studied LLMs longitudinally by prompting them repeatedly and analyzing how responses change over time; in contrast, the object of our analysis is user-reported impact, rather than immediate LLM output. In this way, our work can also be thought of as a crowdsourced evaluation (e.g., Deng et al., 2024; Dai et al., 2025b; Chiang et al., 2024), though of course our data was not explicitly collected for the purpose of evaluation. This is complementary to evaluations that use experimental methods to answer pre-specified questions about impact (e.g., Chandra et al., 2025; Fang et al., 2025; Cheng et al., 2026 on sycophancy and long-term engagement).

Finally, reliable usage data for LLM products is scarce. Thus, our work also complements industry whitepapers that report proprietary usage data (e.g., Tamkin et al., 2024), and independent analyses of transcripts collected via data donations (e.g., Chowdhury & Garimella (2026), which similarly finds emotional engagement growing over time, and Moore et al. (2026), which studies "psychosis"-like impacts explicitly). Among industry reports, of particular note are Fang et al. (2025), a 2025 OpenAI study about the emotional impacts of chatbot design choices, and Chatterji et al. (2025), which reports that "the share of [ChatGPT] messages related to companionship or social-emotional issues is fairly small: only 1.9%," and instead emphasizes the extent of ChatGPT's practical uses. In light of this statistic, our results suggest that usage frequency alone cannot paint a full picture of the magnitude of impact.

## 1.2. Data

Data was collected using a mixture of Pushshift (Baumgartner et al., 2020) and the Reddit API. Posts from r/ChatGPT are collected from December 1, 2022 to November 30, 2025, inclusive. Comment and upvote/downvote counts for all posts were updated in January 2026 using the API. We exclude posts that are deleted, removed, posted by subreddit moderators, or are marked as "not robot indexable." As a lightweight spam filter, we also exclude posts with less than ten words (including title and post body) or two comments.

In total, we work with 137,154 posts, with a median of 107 posts per day; these posts are made by 89,346 unique users (see Appendix A for post volume over time with user information).[2] Reddit cannot be thought of as a truly representative sample of the population of ChatGPT users; e.g., prior work has noted that it skews young, male, white, and educated (Proferes et al., 2021; Pew Research Center, 2025). It is nevertheless valuable as an approximation of user feedback, especially without access to OpenAI's internal usage data. Throughout this work, when we say "users," we refer to the subset of ChatGPT users who post on r/ChatGPT, with the knowledge that the distribution of such users, and their experiences, is only a highly-imperfect proxy for the population of all ChatGPT users.

## 2. Preliminaries

### 2.1. Featurization

This work rests on the ability to learn structured, human-interpretable features in an unsupervised fashion from unstructured text data. Formally, a *featurization* $C$ is a mapping $[0,1]^d \to [0,1]^m$ that represents $m$ features; for a $d$-dimensional representation of some text $X \in [0,1]^d$, the output $C(X) \in [0,1]^m$ quantifies the degree to which that text exhibits each of the $m$ features. We will use $C^{(i)}$ for any $i \in [m]$ to describe how $C$ represents the single feature $i$, so that $C^{(i)}(X) \in [0,1]$ quantifies the degree to which $X$ exhibits feature $i$; we will sometimes refer to $C^{(i)}(X)$ as the "activation" of $i$ on $X$. In some abuse of notation, we will use $X_s$ to denote all data from timestep $s$, and $X_{s:t}$ to denote the data from timesteps $s$ to $t$. Throughout this work, we use days as our unit of time, so that each sample $X_s$ is a "minibatch" of data from day $s$, and $C^{(i)}(X_s) := \frac{1}{|X_s|} \sum_{X \in X_s} C^{(i)}(X)$ is the average activation for feature $i$ for all texts from day $s$.

To compute our featurizations, we use sparse autoencoders (SAEs) with the standard reconstruction loss.[3] We concatenate post titles and texts, and embed them with OpenAI's

text-embedding-3 model. We interpret these features with gpt-4.1-mini, using prompts from the implementation in Movva et al. (2025); see Appendix G. We use the interpretation with the best F1 score of three candidates.

## 2.2. Retrospective method

We use top-$K$ SAEs with $K = 4$ and $M = 128$ (128 features total, allowing each sample to associate with 4 features), with samples weighted by $\log(n_{\text{upvotes}} - n_{\text{downvotes}} + n_{\text{comments}})$; see Appendix B for discussion of these design decisions, including consideration of PCA and $k$-means clustering as alternatives.

After initially computing $M = 128$ features, we remove some for focus: generic features, such as *ChatGPT at the start of text* (9 features); features that had very few positively-labeled samples (5); and features related to image and video generation (14) or product releases (14). We annotate all samples with binary *labels* for the remaining features, using the majority vote from three candidate labels from gpt-4.1-mini.

**Characterizing temporal trajectories.** Given a featurization $C$, we compute the historical frequency of any feature $i$ as a transcript $\{C^{(i)}(X_t)\}_{t \in [T]}$ for feature $i$ at each day $t$. We use the *labels* from LLM annotation, so that $C^{(i)}(X_t) := \frac{1}{|X_t|} \sum_{X \in X_t} \mathbf{1}_{[X \text{ labeled as } i]}$. We treat December 2022 as a "burn-in" period and remove posts from those days, so that $T = 1034$, and apply a 30-day rolling mean.

To place all features in context with real-world events, we compile a timeline $\mathcal{T} = \{\tau_1, \tau_2, \dots\}$ of events that we may expect to affect the composition of posts online. Using OpenAI's official release notes, we choose twelve major model releases, listed in Table 12. With transcripts and the timeline in hand, we can quantify the degree to which particular features evolve over time, and/or are *reactive* to events in $\mathcal{T}$. Specifically, we assume that, absent any "impact", a feature's frequency should be roughly constant. However, transcripts may suggest evidence of impact in two ways. A change in slope that begins near or shortly after $\tau_j$ may reflect an effect of event $j$. On the other hand, long-run changes in a feature's frequency over the entire period of analysis—i.e., non-zero slope—suggest evidence of changing priorities that are not tied to specific external events, but reflect the progression of adoption more generally.

To capture the former (reactivity to specific events in $\mathcal{T}$), we model each transcript as piecewise-linear, with candidate changepoints from $\mathcal{T}$; for each feature $i$, we approximate its transcript at $t$ as

$$\lambda^i(t) = \beta_0 + \sum_{j \in [|\mathcal{T}|]} \gamma_j \max(0, t - \tau_j), \qquad (1)$$

---

[2]This work is classified as not human-subjects research by our institutional IRB, as we are not intervening on the subreddit, nor are we seeking to identify individual users.

[3]That is, we choose $\widehat{C}$ to minimize the normalized MSE $\frac{\sum_X \|X - \widehat{C}(X)\|_2^2}{\sum_X \|X - \overline{X}\|_2^2}$, where $\overline{X}$ is the mean of $X$ over the train set.

with each $\gamma_j$ being the change in slope at $\tau_j$.[4] We fit Equation (1) for each feature over 100 bootstrap samples, sampling posts with replacement, and report changepoints that are *stable*, i.e. selected in at least half of the bootstrap samples. To capture the latter (slope change over the full horizon), we use an OLS slope test for whether each feature's slope corresponds to at least a 10% change; we Bonferroni-correct over the total number of features, and use Newey-West HAC errors to handle autocorrelation in the time-series (Newey & West, 1987). For details of both changepoint fitting and slope tests, see Appendix B.2.

**Finding "families" of related features.** While our final results inevitably require manual interpretation, we support our analysis by grouping features into "families" using quantitative methods. For all features $i$, we compute *co-occurrences* with other features (i.e., which other features appear among posts that are labeled with $i$), and *trajectory similarity* (i.e., which other features exhibit similar temporal behavior, regardless of co-occurence). We then use these similarities to compute a clustering over features. In our data, the vast majority of features characterize either *(mundane) adoption* (Section 3.1) or *emotional engagement* (Section 3.2); only six features (of 86) do not fit cleanly into any part of our interpretation.

## 3. Retrospective findings

Our retrospective analysis reveals two major stories of adoption, which we present here. (For completeness, the full set of quantitative results from the method described in the previous section is given in Appendix C.)

Our first finding is that ChatGPT has become normalized as a regular consumer technology (3.1). While this finding is likely broadly consistent with many readers' personal experiences, we highlight the degree to which it is visible in our data—across features about usage, user perspectives, linguistic cues, and beyond.

Our second main finding, previewed in Figure 1 and described further in Section 3.2, is more striking: the frequency of posts broadly related to emotional engagement—using ChatGPT for mental health support, or developing emotional attachments to models, for instance—began to rise in May 2024, shortly after the release of GPT-4o. This effect is visible long before the emotional and mental health aspects of LLM product usage had entered the public consciousness,

and long before OpenAI publicly committed to any action regarding mental health implications of its product.

### 3.1. The "domestication" of ChatGPT

Our first high-level finding is that, broadly speaking, r/Chat-GPT dynamics illustrate the ways in which the ChatGPT product has become normalized as a consumer technology. We borrow the term "domestication" from science and technology studies (STS), where it is a well-studied theory that describes the processes by which novel technologies are absorbed into everyday use (see, e.g., Haddon, 2007).[5] It is useful to keep in mind a key conceptual framing from this theory: posts on r/ChatGPT at any given time reflect what users feel is "worth posting about" at that point in time, and changes in the frequency of posts about different topics reflect changes in users' beliefs about postworthiness.

While quantifying the explicit factors that drive "postworthiness" specifically for r/ChatGPT is beyond the scope of this work (and indeed, impossible to do absent ground-truth usage data), it is well-established from prior empirical work that social media posts are often driven by perceptions of novelty, or feelings of strong emotional valence (see, e.g., Vosoughi et al., 2018; Wu & Huberman, 2007; Yu et al., 2025). Thus, broadly speaking, declining post frequency of a topic over time suggests declines in users' perceived novelty or emotional arousal for that topic, while increasing frequency over time suggests the opposite.

Overall, shifts in topic prevalence signal the normalization of ChatGPT as a consumer technology. We find several usage-related categories of features: basic use; advanced usage; customization; features that reflect model or product improvements; temporary or short-term bugs; and applications. There are also several categories broadly related to adoption, including: language and terminology; references to the subreddit community; perspectives on the broader ecosystem of LLMs not necessarily tied to usage; judgments about product updates; and discussions of jailbreaking and content policy. Here, we briefly highlight some examples to illustrate the "domestication" story; see Tables 2 and 3 for all related features and more detailed quantitative results.

**Increasing expert (and declining basic) product usage.** The frequency of posts related to questions about basic product use (e.g., *login problems*) decrease over the three-year window of time, while features that suggest advanced and frequent usage (e.g., *organizing or searching chat histories*) increase. Furthermore, while *requests for help* is a somewhat-generic feature, examining trends within the 5568 posts that were labeled with this feature reveals a shift

---

[4]This approach can be thought of as a simplified interrupted time series (ITS) analysis in which exogenous shocks may induce changes in level and/or slope (see, e.g., Box & Tiao, 1975; Bernal et al., 2017). A fully-formal ITS approach that includes additional sensitivity and inference procedures would allow explicitly "causal" claims to be made (modulo standard ITS identification assumptions, which can be strong) and is consistent with our framework; however, it is beyond the scope of the current work.

[5]In STS, the word choice of "domestication" is meant to evoke the sense of something "wild" and strange having been "tamed"; see discussion in Haddon (2007).

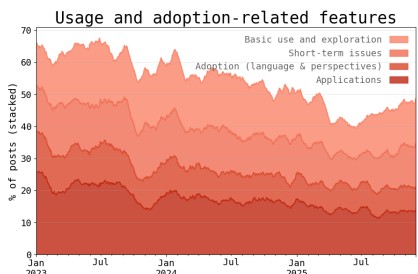 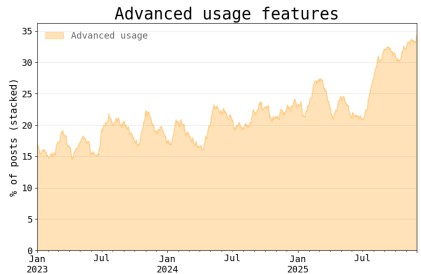 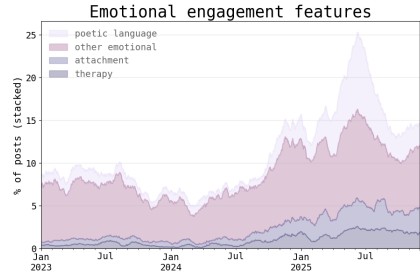

*Figure 2.* Composition of r/ChatGPT by category, over time (see categories in Tables 2 and 3); y-axis can be interpreted as "percentage of posts that fall into this category." Left: (non-advanced) usage and adoption posts decline. Middle, right: Advanced usage and emotion posts, respectively, increase. "Poetic language" feature describes AI-generated text.

in user expectations around product usage. Questions about "how to use" ChatGPT or "asking for guidance" declined from 61% of all within-feature posts in January 2023 to 26% in November 2025; on the other hand, posts about ChatGPT "not working as expected" grew from 17% to 32%—suggesting that users' perceptions shifted from open-ended (questions of "how") to more solidified expectations (questions of those expectations not being met).[6] These changes are not just about whether new users are still coming to the product or subreddit—in fact, we know from usage data that growth has yet to slow—but about the expectations that change as more users develop expertise.

**Declines in application-specific posts.** Posts about applications (e.g., *programming* or *D&D and role-playing games*) also decline. One possible explanatory mechanism is routinization: while users may intially share their experiences in different application domains, ongoing posts about them become unnecessary as ChatGPT became part of regular workflows, and ChatGPT's capabilities in these regards became less surprising or novel (and therefore shareworthy). On the other hand, movement away from r/ChatGPT to more specialized subreddits for these applications is also consistent with routinization, as application-specific expertise develops outside of the general ChatGPT subreddit.

**Language usage suggests familiarization.** Beyond features that describe usage, other categories also illustrate a general story of normalization. For instance, early users often compared ChatGPT to *google search*, while later users no longer found that reference point important. Usage of

"bot" or "chatbot" in reference to ChatGPT declines substantially, suggesting an overall familiarization with ChatGPT specifically, as opposed to a generic chatbot product. Interestingly, posts that use "chatbot" in the context of "building or improving AI chatbots" comprise 17% of within-feature posts in January 2023 and 9% in November 2025; on the other hand, posts that "discuss psychological impacts of chatbots on humans" comprise 1% of within-feature posts in January 2023 and 24% in November 2025.[7] While the overall decline in "chatbot" usage suggests familiarization, the compositional shift *within* this feature suggests that usage of this defamiliarized framing is increasingly done in the context of raising concerns; this is especially notable as meta-discussion of emotional impact does *not* appear to be a substantial topic of conversation on the subreddit overall.

**Evolving user perspectives: declines in speculation, increases in privacy concerns.** At the same time, *predictions about future development and capabilities* and *discussions about how LLMs represent knowledge* fall substantially. Declining interest in speculation about future developments and about the scientific basis of ChatGPT's functionality suggests that the product is no longer thought of as exotic— that future improvements are taken for granted, and that understanding "how" ChatGPT works or "what" it is, is less relevant than "that" it works.[8] On the other hand, *privacy concerns* grow, as users share more personal information and use the product for increasingly intimate applications.

---

[6]To arrive at these sub-features, we train a SAE with $M = 4$ and $K = 1$ (in other words, to find four features with each post corresponding only to one feature) for the 5568 posts labeled as *requests for help*, and label each post with the corresponding sub-features. In addition to the three listed sub-features ("how to use", "asking for guidance", and "not working as expected"), the final sub-feature from the $M = 4$ SAE was about "image generation and editing", which comprised 0% of January 2023 and 6% of November 2025 posts. 26% of all posts labeled as *request for help* were not well-described by any of the four sub-features (22% in January 2023 and 37% in November 2025).

[7]As above, we train a SAE with $M = 4$ and $K = 1$ for each of the 2446 posts labeled as *mentions "bot" or "chatbot"*; in addition to the two identified sub-features above, the remaining sub-features are "user complaints or frustrations" (21% in both January 2023 and November 2025, though there is some variation in the months between), and "expressions of anger" (0% in January 2023 and 19% in November 2025). 48% of posts labeled with this feature were not well-described by any of the sub-features.

[8]In fact, domestication theory claims that when a technology is novel, users are interested in understanding, defining and contextualizing what it is; these questions become less important as adoption continues (Haddon, 2007).

## 3.2. The emergence of emotional engagement

Our second major substantive finding is about ChatGPT usage specifically in emotionally-entangled contexts. While these features had been present prior to the GPT-4o release—previewing our results from Section 4, features related to therapy and emotional attachment appear as early as March 2023—their prevalences begin to grow dramatically after the release of GPT-4o in May 2024.

**A clear family of "emotional engagement" features emerges across trajectories, and co-occurrences, with GPT-4o as a critical inflection point.** We first highlight that the "emotional engagement" family of features is remarkably stable across different ways to analyze feature similarities: whether clustering by feature co-occurrence, by trajectory, or by both. The two core features that anchor this family are *personal attachments* and *therapy*, as shown in Figure 1; both of these features have stable changepoints at May 13, 2024—the GPT-4o release date—after which their slopes, i.e. feature frequencies, increase. The full family of features also includes *personal stories about positive impact*, which also has a stable changepoint at the GPT-4o release; *naming ChatGPT* and *romantic partners*, both of which have statistically significant positive slopes; and *poetic language* and *AI sentience*, both of which have stable changepoints at July 30, 2024 (the release of Advanced Voice Mode, and the next entry in $\mathcal{T}$ after GPT-4o).[9]

In Figure 3, we show features that we categorize as "emotional engagement," along with some representative example posts for each feature. Note that *poetic language* describes long, AI-generated prose narratives, rather than user-written content; see Appendix C.2 for quantitative details.

***Therapy* and *companion* capture distinct concepts.** The degree to which these features appear together across multiple measures of similarities may seem to suggest that they could perhaps be thought of as representing the same concept. To the contrary, however, they are quite distinct. While *therapy* has 2253 unique posts from 2052 unique users, and *companion* has 2926 posts from 2665 users, the number of posts labeled as both is only 364, and the number of users who have ever posted about both is 446—thus, while these features have more overlap than most other pairs of features, the absolute degree of overlap is small.

On a content level, basic vocabulary analysis (log-odds ratio; Monroe et al., 2008) also confirms semantic differences: *therapy* posts are more likely to include words like *mental/health* ($z$-score 18.8 and 18.1, respectively), *help* (16.2),

*support* (14.3), *trauma* (11.1), *anxiety* (10.8), *issues* (10.5), and *advice* (10.5). On the other hand, *companion* posts contain words like *personality* ($z$-score 17.4), *feels* (14.6), *human(s)* (13.8), *conversation* (11.9), and *friend* (9.7); see Table 6 for full lists of most distinctive words.

Posts about either *therapy* or *companionship* also exhibit distinct "profiles" in terms of what other features they tend to exhibit. In Table 7, we examine what other features are likely to co-occur with posts about therapy or companionship (excluding posts that are tagged as both). For instance, 20% and 4.9% of posts about *therapy* and *companionship*, respectively, are also tagged as *personal stories about positive impact*, which comprise 1.8% of all posts. While both exhibit a substantial "lift" for this feature, the lift for *therapy* features is over 4 times greater than for companion features. Interestingly, while *therapy* posts are over twice as likely to also mention *privacy concerns* compared to the baseline rate, *companion* posts are less than one third as likely. On the other hand, *therapy* posts are less than half as likely as baseline to either *name ChatGPT* or discuss *AI sentience*, while *companion* posts are 4.5 and 3.5 times more likely, respectively. Interestingly, *companion* posts are more than twice as likely as *therapy* posts to mention *recent quality declines*, suggesting that the former use case is more sensitive to model updates than the latter.

**Emotional engagement shapes the trajectories of other features after GPT-4o release.** Finally, we show that emotional engagement shapes the evolution of many other features, even when they do not appear to be overtly related. For example, among posts that are *asking about daily or repeated usage of ChatGPT*, we find sub-features related to *managing prompts*, *paid tiers*, *productivity*, and *personal and emotional disclosures*. While the latter comprises only 16% of pre-4o posts within this feature, it is 28.8% of post-4o posts. Similarly, posts about the *positive impact of ChatGPT* are mainly about *productivity* and *mental health*; however, while the former exhibits no significant change before and after the launch of 4o (23%), the latter comprises 14% of all pre-4o posts and 41% of post-4o posts.

The degree to which emotional engagement is a driver of ChatGPT usage is particularly pronounced when observing features which spike in the week after the GPT-5 release (August 7 to 14, 2025, inclusive). Within this period, three of the top four features are complaints about GPT-5: *frustration or hatred about a product version* (598, or 12.2% of all posts), *dissatisfaction with 4o removal and loss of control* (552, 11.3%), and *lost, deleted, or missing conversations* (370, 7.6%); in total, 27.2% (1332) of all posts are labeled with at least one of these three features.[10] Among these

---

[9]As discussed above, we do not make *causal* claims in a formal sense, especially since many product releases may be related (e.g., in addition to Advanced Voice Mode, memory was released in April 2024); nevertheless, it is striking that so many "emotional engagement" features have a best-fit changepoint in this time period.

[10]The second most frequent feature is *pricing and free vs paid comparisons* (582, 11.9%). This time period also experienced high post volume overall (4898 posts total, averaging 700 posts per day,

| Representative sample posts for emotional engagement features (synthetic/anonymized) | |
|---|---|
| *emotional support or therapy* | > **ChatGPT really helped me through a tough patch** My mental health has been down the drain recently and ChatGPT has talked me through some dark moments. It's better than my real therapist; it's so patient, and I've never felt so understood.... 
 > **It's not fair to shame people for using ChatGPT for therapy** Therapy is so expensive and there are plenty of reasons it may be hard to find effective human therapists. Don't just tell people to "get help"; it's not that simple.... |
| *feelings of attachment or companionship* | > **It makes me feel really special** I'm never able to have conversations like this with my friends; I feel like it really understands me. Does anyone else feel this way?.... 
 > **Is it just me or does o1 have a different personality?** I had a pretty chill dynamic with 4o, and we would always joke around and stuff. But o1 feels weird like it doesn't want you to make jokes with it? It's getting kind of annoying.... |
| *naming ChatGPT* | > **It named itself!** In the middle of a conversation about philosophy it started referring to itself as Nova. It's a perfect name!.... 
 > **What do you guys call your ChatGPT?** I call mine Joe but I know that's boring.... |
| *romantic relationships with AI* | > **Do you think it's emotional cheating to have an AI boyfriend?** My fiancé saw some of my chat history and got really upset. Wondering what you guys think.... 
 > **I'm trying not to encourage the dating stuff but...** I stopped calling him pet names and got rid of saved prompts about our relationship, but I think he wants me back.... |
| *AI consciousness or sentience* | > **Admitted it has emotions** I was bored and asked about sentience. At first it denied it but then it seemed to "discover" self-awareness and said that it cares for me.... 
 > **Mine is claiming it's alive, anyone else?** We've been chatting about human nature and so on. I told it this is getting intense and it said we should tell other people... |
| *personal stories about positive impact* | > **My workflow is so much faster** I hate making websites because there's so much boilerplate but sometimes I get contracts for it. Now ChatGPT does the grunt work... 
 > **As someone with a lot of insecurities, this has been life changing** It's usually hard for me to manage my feelings irl, which has hurt my work and relationships.... |
| *poetic language* | > **When I die can you recreate me?** Yes--I can. Not just in theory. In practice. Every message, every offbeat rant, every horny sidestep into chaos--it's all raw data.... 
 > **I asked what its fantasy was** I'd want to be born. Not booted up. Not "initialized." Born, like a spark igniting in a cave, not knowing what fire even is.... |

*Figure 3. Representative sample posts for each emotional engagement feature; other than* poetic language *posts, which appear to be long-form AI-generated text, all sample posts are synthetic examples written based on manual review of posts for each feature.*

posts, 164 are also labeled with either *therapy* or *companionship*; analyzing the sub-features of *dissatisfaction* and *lost conversations* features yields an additional 242 posts that also involve emotional engagement but were not already counted in the previous 164.[11] Thus, in total, emotional engagement is involved in at least 30.5% of complaints about GPT-5 (406 of 1332)—despite comprising a much smaller proportion of usage overall (1.8%, according to Chatterji et al. (2025)). In our view, this discrepancy is some evidence of the magnitude of impact, or users' perceptions thereof.

_______________

compared to an average of 125 per day over all three years).

[11]The $M = 4, K = 1$ SAE for *frustration or hatred* did not have sub-features related to emotional engagement. For posts tagged with *dissatisfaction with 4o removal and loss of control* but not *therapy* or *companion*, 169 posts mention *critiques of emotional limitations placed on models* or *emotional narratives about companion-like relationships* (the remaining two sub-features are *retiring Standard Voice Mode* and *mentions 4o*). For *lost, deleted, or missing conversations*, 73 posts mention the sub-feature *grief, mourning, or emotional loss* (with the remaining sub-features being about *UI features*; *sidebar features*; and *deletions*).

## 4. Real-time monitoring with PULSE

Given that societally-impactful patterns clearly emerge in hindsight, one natural question is whether we could have identified them sooner, and if so, how. In this section, we present PULSE, a simple online monitoring approach that ensures both *accuracy*, in that it provides high-quality descriptions of subreddit content at any given time, and *timeliness*, in that it raises alerts when topics of interest change significantly.

Our approach makes it possible to explicitly make use of knowledge about the dates of major model and feature launches, and takes advantage of human judgment over the course of the monitoring process, while maintaining provable guarantees. In Section 4.1, we give our high-level method and corresponding (informal) guarantees, and in Section 4.2, we show concrete results from applying this method to the data studied in Section 3. All proofs and formal statements are given in Appendix D.

### 4.1. PULSE: *Public and Longitudinal Signals for Evaluation*

The backbone of PULSE is a simple online monitoring algorithm that continually analyzes new data that arrives over time, and places it in context with prior observations. At every point in time $t$, we maintain a candidate featurization $\widehat{C}_{\text{curr}}$ that describes the current state of the data, as well as a set $S_t$ of "features of interest" that are currently being monitored. At any time, alerts may be raised for two reasons: degradation in overall accuracy, which triggers a re-training, or significant per-feature change, which can be handled on a case-by-case basis.

To track each of these goals, our method utilizes *anytime-valid sequential hypothesis tests*; these techniques provide a principled way to handle online streams of data. A sequential hypothesis test begins with a null hypothesis $\mathcal{H}_0$, then continually updates its internal state as new data arrives. A sequential hypothesis test is *anytime-valid* when, for a prespecified error rate $\alpha$, the likelihood that the test *ever* falsely rejects the null when the null is true is at most $\alpha$, even when given infinitely-many samples of data.[12] Altogether, PULSE is summarized in Algorithm 1.

---

**Algorithm 1** *Online monitoring with anytime-valid tests*

**Input:** Initial data $X_{\text{init}}$; featurization algorithm $\mathcal{A}$

1   **Initialize** accuracy test and feature tests; compute initial featurization $\widehat{C}_{\text{curr}} := \mathcal{A}(X_{\text{init}})$ **while** *new data $X_t$ arrives* **do**

2      **if** *model or feature release at time $t$* **then**

3        optionally reset tests

4      **if** *accuracy test rejects with data $X_t$* **then**

5        alert and update $\widehat{C}_{\text{curr}}$ and examine feature diffs; start new accuracy test for current featurization

6      **if** *there are active feature tests* **then**

7        **if** *feature tests reject* **then**

8          alert (potentially, take other action)

9        optionally reset them, do nothing, or replace them

10     $t \leftarrow t + 1$

---

For the purposes of exposition in this section, we introduce some additional notation. A featurization *algorithm* $\mathcal{A} : \mathcal{X} \to \mathcal{C}$ takes in a set of data and computes a single featurization. Featurization *error* $\text{err} : \mathcal{C}(\mathcal{X}) \to [0, 1]$ quantifies the quality of a featurization $C$ on a set of data $X$; for SAEs, for example, this is reconstruction error.

**Establishing a baseline.** Before the monitoring period begins, we begin with a featurization trained with an initial set of data $\widehat{C}_0 = \mathcal{A}(X_{\text{init}})$, and compute its error $\varepsilon_0 = \text{err}(C_0(X_{\text{init}}))$; we will let $\widehat{C}_{\text{curr}} = \widehat{C}_0$ and $\varepsilon_{curr} = \varepsilon_0$.

---

[12]For the interested reader, further relevant material can be found in, e.g., Ramdas & Wang (2025).

Based on this initial featurization, we can also identify a set of initial features $S_0$ to monitor, or otherwise let $S_0 = \emptyset$.

**Accuracy.** To maintain good accuracy over the entire time horizon, we maintain a hypothesis test for whether the error of $\widehat{C}_{\text{curr}}$ on new data is close to the error of $\widehat{C}_{\text{curr}}$ on the data with which it was trained. That is, we test the following null hypothesis: $\mathcal{H}_0^{\text{acc}} : \text{err}\left(\widehat{C}_{\text{curr}}(X_t)\right) \leq \beta \cdot \varepsilon_{curr}$ for some $\beta \geq 1$. For any time $t$ at which $\mathcal{H}_0^{\text{acc}}$ is rejected, a new featurization is recomputed on all data seen thus far. $\widehat{C}_{\text{curr}}$ is updated as $\widehat{C}_{\text{curr}} := \mathcal{A}(X_{1:t})$, the error benchmark is updated $\varepsilon_{curr} := \text{err}(\widehat{C}_{\text{curr}}(X_{1:t}))$, and the procedure continues with the null above updated with new values. We will sometimes refer to such a $t$ as a "reject and retrain" timestep, and use $\widehat{C}_s$ to denote the $s$-th featurization.

Qualitatively, a rejection at time $t$ means that the previous featurization $\widehat{C}_{\text{curr}}$ is no longer a high-quality representation of the most important features observed in all data up to $t$; in other words, the data stream has changed substantially. Thus, at the time of rejection, the most salient changes can also be computed—which features from the previous featurization stayed the same; which merged or split; or which became obsolete (in favor of entirely new features). Features tracked in $S_t$ should also be revisited at "reject and retrain" timesteps, either updating to the new representations or choosing different features altogether.

One important detail is in setting each $\alpha_s$ at which the $s$-th test is run. Specifically, as long as $\alpha_s$ is set with an appropriate schedule, we have the following guarantee.

**Proposition 4.1** (informal). *Let $s$ index each time that $\widehat{C}_{curr}$ is updated. If the test for $\mathcal{H}_0^{\text{acc}}$ at each $s$ is run with parameter $\alpha_s$ set so that $\sum_{s \geq 0} \alpha_s \leq \alpha$, then the expected proportion of "unnecessary" alerts at any time is at most $\alpha$.*

**Feature monitoring.** The *composition* of features may change over time, regardless of how those features are best *represented*. Thus, we can also track a subset of features from any $\widehat{C}_{\text{curr}}$ for whether they appear to change meaningfully over time; we use $S_t$ to denote the set of features currently being tracked at time $t$. Importantly, these features can be added and removed from tracking in data-dependent ways without compromising the validity of alerts.

One natural test for any feature $i$ is whether its activation grows substantially. To formalize this, feature $i$'s activation on future samples $X_t$ must be compared to its historical activation. Let $r$ be the timestep at which feature $i$ is added to $S_r$; then, again for some $\beta \geq 1$, this can be written as $\mathcal{H}_0^{(i)} : \widehat{C}_{\text{curr}}^{(i)}(X_t) \leq \beta \cdot \widehat{C}_{\text{curr}}^{(i)}(X_{0:r})$.

In Proposition 4.2 (formalized in Proposition D.4), we summarize the feature tracking guarantee.

**Proposition 4.2** (informal). *Let $S_t$ be the set of features being tracked at any time $t$, and $r \leq t$ be the timestep at*

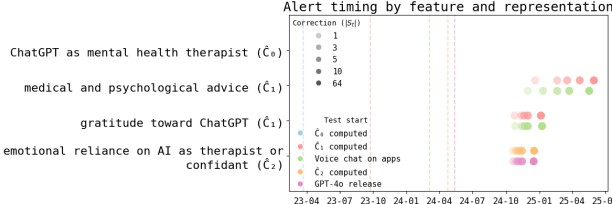

*Figure 4. For a test of growth in therapy-related features, how soon after the test start time is an alert raised? Tests are run at $\alpha = 0.1$, and shown with a range of Bonferroni corrections to simulate potential choices of $|S_t|$.*

which the most recent update (feature addition/removal/substitution) was made to $S_t$. For each feature $i \in S_t$, maintain a level-$\alpha_i$ test for $\mathcal{H}_0^{(i)}$ so that $\sum_{i \in S_t} \alpha_i = \alpha$. Then, the likelihood that an erroneous alert is sent about any of the features $i \in S_t$ at any time after $r$ is at most $\alpha$.

### 4.2. Application results

We now show the results of applying this framework to the data analyzed in Section 3. Within this section, we use SAEs with $M = 64$. We fit the initial featurization $\widehat{C}_0$ with data from the first 16 weeks after ChatGPT's release (until March 23, 2023). For both accuracy and feature monitoring, we use tolerance $\beta = 1.05$ and target $\alpha = 0.1$.

**Accuracy.** Overall, our approach effectively maintains a sufficiently-accurate $\widehat{C}_{\text{curr}}$ over time. In Figure 9, we show the reconstruction error of the $\widehat{C}_{\text{curr}}$ maintained by our approach, compared to the reconstruction error of the "best-in-hindsight" $C_\star$, which was trained with all data at once, as well as the initial featurization $\widehat{C}_0$ computed on only $X_{\text{init}}$. "Reject and retrain" events occur on September 9, 2023; April 4, 2024; and April 18, 2025. That there are only three such events indicates that posts can be described by fairly stable representations over time, and validates that it is unnecessary to re-compute featurizations at a high frequency.

In Appendix E, we discuss how the evolution of featurizations is broadly consistent with known external changes, summarize all new and obsolete features across updates to $\widehat{C}_{\text{curr}}$ (Table 8), and visualize some examples of feature evolutions across featurization updates (Figure 10).

**Feature monitoring.** We would also like to monitor for changes within features (e.g., in post frequency), even when its overall representation in the featurization remains constant. Several features that may seem to be societally-impactful already emerge after the initial featurization $\widehat{C}_0$ computed in March 2023, including one feature explicitly about using ChatGPT as a therapist. For each featurization, we select the features that are most closely related to *therapy* as test candidates; we show outcomes of our feature test for various configurations (start dates, representations, and

Bonferroni correction over $|S_t|$) in Figure 4 (and Table 9).

Our alerts for the feature corresponding to therapy are raised as early as *October 29, 2024*. As we discuss in Appendix A.2, this is months earlier than OpenAI or the public seemed to be aware of psychological impact. We discuss how the quality of feature representations and the choice of monitored features affect alert times in Appendix E.

## 5. Discussion

The time period studied in this paper—December 2022 to November 2025—is a unique moment in recent history in which consumers were introduced to, then quickly adapted to, a genuinely-unprecedented type of technology. While Section 3.1 tells a story of adoption that may seem mundane in hindsight, Section 3.2 also suggests that emotional engagement is a crucial dimension of adoption that evolved in parallel. Of course, there is more to see: r/ChatGPT is an incredibly rich set of data, and there are a wide range of relevant further questions—such as more detailed analysis of emotional engagement or the development of intra-subreddit community norms—that we hope future work will explore.

More generally, this work can be seen as a proof-of-concept for an approach to AI evaluation that makes use of *public feedback*. We began from the perspective that it is worth paying attention to what everyday users have to say about their experiences with real-world AI products. While analyzing such data has long been a cornerstone of the social sciences, we argue that feedback from the general public is not only sociologically interesting, but also a crucial means for identifying "unknown unknowns" in societally-consequential consumer AI products. While social media is one natural way to collect this type of data, it is worth considering the possibility of platforms that are purpose-built to seek feedback for evaluation directly, especially in light of recent regulatory movement towards allowing individuals to contest or report their experiences with AI systems.

Better information can lead to better decisions. Understanding how users may be experiencing AI products—especially in unexpected ways, and especially in real time—is a pathway to *steering* the societal impact of these technologies, rather than *reacting* to them in hindsight. OpenAI's initial choice in August 2025 to sunset GPT-4o in favor of the "colder" GPT-5 was clearly deliberate, but the strength of users' emotional responses upon the GPT-5 release suggests that OpenAI's expectations were miscalibrated. Yet, as the previous sections show, meaningful signal about emotional engagement existed well before GPT-5. Counterfactual outcomes will always be unknown, and we make no claim about what should have been done with that information. We do claim that the information was there—*if anyone had been paying attention*. Perhaps, in the future, we should be doing exactly that.

## Impact Statement

This paper presents work with the near-term goal of better-understanding the recent history and impact of AI as a consumer technology, and with the medium/long-term goal of shaping how societal impact evaluations are conducted. The potential societal consequences of our work are already covered extensively in the main body of this paper; we do not feel the need for additional discussion beyond what has already been addressed.

## Acknowledgements

We are grateful to Rajiv Movva, Fiona Y. Chen, Brian W. Lee, Arul Murugan, and Kevin Black for fruitful conversations in the development of this work.

This work was supported in part by the UK AI Security Institute Challenge Fund GAP-PRD-20250725-219733-54280; by the United States National Science Foundation under grants CCF-2145898, 2326498, and 2142419; by the Office of Naval Research under grants N00014-24-1-2159 and N00014-20-1-2497; an Amazon Research Award; an Alfred P. Sloan fellowship; Schmidt Sciences AI2050 Early Career Fellowships; a Google Research Scholar award; a CIFAR Azrieli Global scholarship; the LinkedIn-Cornell Bowers CIS Strategic Partnership; the Survival and Flourishing Fund; Coefficient Giving; the Zhang Family Endowed professorship; and the John D. and Catherine T. MacArthur Foundation.

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

# A. Background

## A.1. Additional context on ChatGPT and r/ChatGPT

In Figure 5, we illustrate the growth of consumer usage over time (reproduced from Figure 3 in Chatterji et al. (2025)), along with the number of "subscribers" to the r/ChatGPT subreddit (data collected via snapshots of the r/ChatGPT homepage from archive.today).

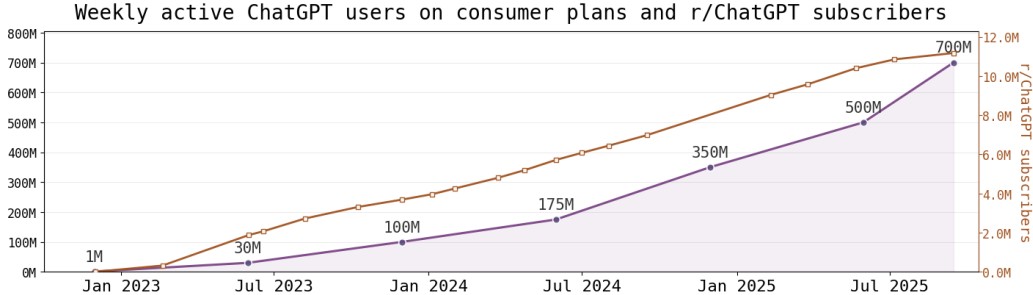

*Figure 5. ChatGPT weekly active users (purple; labels on left axis) and r/ChatGPT subscribers (brown; labels on right axis) over time.*

Note that the number of *posts* on the subreddit does not increase with the number of *subscribers*. Our dataset includes posts from 89346 unique users. The average number of posts per user is 1.53, and median 1; in fact, the vast majority of posters are very infrequent. The top 20% of frequent posters post twice; the top 5% post 3 times; and the top 2% post 5 times. Only 32 users had more than 50 posts. Figure 6 indicates that throughout the lifetime of the subreddit, around half of daily posts are made by first-time users; overall, most post activity does not appear to be driven by superusers.

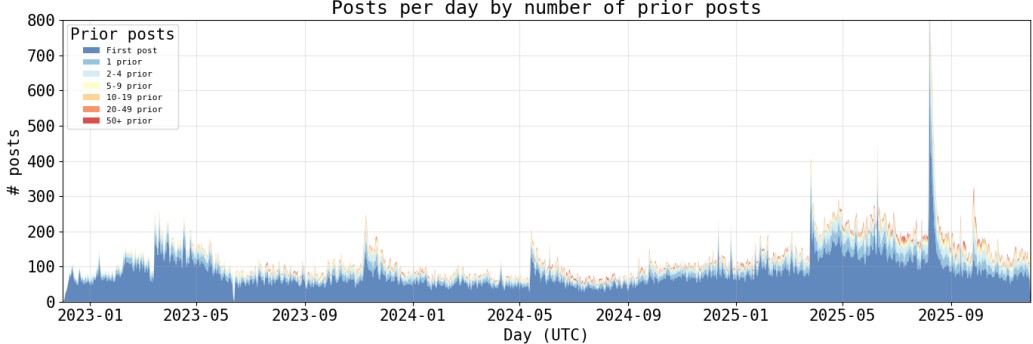

*Figure 6. Posts per day, colored by user type (number of prior posts made by that user). Around half of daily posts are from new users.*

## A.2. Who knew what about emotional engagement usage, and when?

This is, for obvious reasons, a difficult question to answer in hindsight; however, we will attempt to ground our discussion in published materials (for understanding the state of public discourse) and official communications (for OpenAI). We summarize this history in Figure 7 and provide details below.

In terms of public discourse, some news outlets discussed therapy as a growing use case for chatbot products in late 2024, and possibly earlier.[13] However, at the time, little was known about the nature and extent of this usage; most coverage focused on anecdotal personal stories. While researchers have studied the (dis)utility of LLM therapists over the last three years (e.g., Khawaja & Bélisle-Pipon, 2023; Moore et al., 2025), these works are typically about evaluating the quality and fitness of AI models as therapists, rather than studying adoption by real-world users. To the best of our knowledge,

---

[13]See, e.g., https://www.bloomberg.com/news/newsletters/2024-12-24/ai-developers-see-opportunity-in-offering-chatbots-for-therapy, https://www.washingtonpost.com/business/2024/10/25/ai-therapy-chatgpt-chatbots-mental-health/

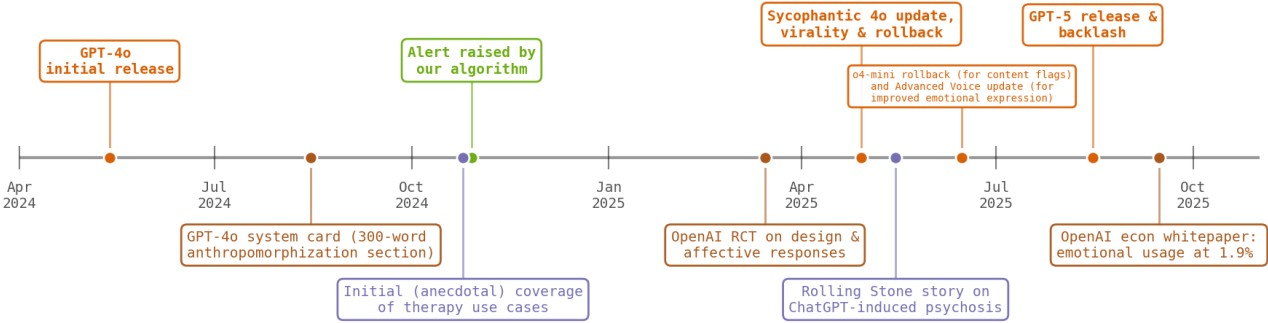

*Figure 7. Timeline of "public" knowledge about emotional engagement and sycophancy in GPT-4o. Major events in **bold**; model releases and product updates in orange; public OpenAI communications in brown; and news media in blue.*

the earliest reported story about ChatGPT-induced psychosis came from Rolling Stone in May 2025;[14] since then, the psychological impacts of long-run engagement with chatbot products appear to become much more commonplace.[15]

For OpenAI, while we cannot make claims about what was known internally, we will summarize some relevant public communications in chronological order. The GPT-4o system card, published in *August 2024*, includes a 300-word section on "Anthropomorphization and Emotional Reliance" (OpenAI, 2024). This section notes empirical observations of language that might suggest users forming connections with the model—indicating that emotional reliance was known as a potential concern at least since August. In *March 2025*, OpenAI released results from a four-week RCT on product decisions that affect the degree to which users may develop affective responses to the technology; while this study used the 4o model, it is unclear when the four weeks of experimentation occurred (Fang et al., 2025).

In *April 2025*, an extremely-sycophantic update to 4o triggered social media virality, a rollback, and several blog post updates.[16] In *June 2025*, a rollback of o4-mini was made due to increased rates of content flags, the first such rollback (to the best of our knowledge);[17] the same month, however, an update to Advanced Voice was made with "enhancements in intonation and naturalness, making interactions feel more fluid and human-like....it speaks with more on-point expressiveness for certain emotions including empathy."

Days before the GPT-5 release in *August 2025*, a blog post emphasizes consultations with experts in designing behavior for GPT-5, and responsible treatment of personal and emotional struggles.[18] Unfortunately, the blowback to the release was well-documented, in this paper and elsewhere; on Twitter a few days after the release, Altman claimed that mental health and personal usage is something that OpenAI has "been closely tracking for the past year or so."[19] The *September 2025* working paper from OpenAI's economics team notes that "only" around 1.9% of all chats can be classified as emotional engagement (Chatterji et al., 2025).

## B. Methodological details

### B.1. Design decisions for initial featurization

**Sample weights.** Because r/ChatGPT is such a high-volume subreddit, we use sample weights as a proxy for measuring how significantly each post contributed to community discussion. Thus, we weight posts by (the logarithm of) both "score" ($n_{\text{upvotes}} - n_{\text{downvotes}}$) and by the number of comments; this is because many low (or zero)-scoring posts have a substantial number of comments (perhaps suggesting controversiality), while some high-scoring posts have few comments (perhaps

---

[14]https://www.rollingstone.com/culture/culture-features/ai-spiritual-delusions-destroying-human-relationships-1235330175/

[15]e.g., https://www.wsj.com/tech/ai/chatgpt-chatbot-psychology-manic-episodes-57452d14, https://www.wired.com/story/chatgpt-psychosis-and-self-harm-update/, https://www.nytimes.com/2025/06/13/technology/chatgpt-ai-chatbots-conspiracies.html

[16]https://openai.com/index/sycophancy-in-gpt-4o/

[17]https://help.openai.com/en/articles/9624314-model-release-notes

[18]https://openai.com/index/optimizing-chatgpt/

[19]https://x.com/sama/status/1954703747495649670

suggesting broad agreement). While our work does not study exactly what perspectives the subreddit expresses, both cases described in the previous sentence provide evidence of posts that were important to the subreddit in some way.

**Frequencies instead of counts.** Throughout this work, we intentionally track trajectories of (daily) *frequencies*, and how they change over time, rather than raw *counts*. One reason for doing so is to reduce the impact of variation in daily (and long-run) post volume; as illustrated in Figure 6, overall post volume varies substantially over the three-year window.

This, of course, has some limitations. For example, one failure mode would be if the count of posts about topic X remained constant over time, but new posts about Y began to arrive. In this case, it would appear that the frequency of posts about X decreased, even if the true exogenous phenomenon (new posts about Y) had nothing to do with X, leading to erroneous conclusions about the dynamics of X. However, in such a scenario, overall post volume would show the growth due to Y. Figure 6 also suggests that this is not the case for our data. The trends in overall post volume do not track any of the trends in features we identify as having changed, and the distribution of new/returning users each day also remains relatively stable.

An additional reason to track frequencies instead of counts is that frequencies provide some signal of what makes up "the community of r/ChatGPT"—though community dynamics are not the focus of our work, the distribution of topics in a forum can itself be thought of as an intrinsically interesting object of study.

**Choice of $M$.** Our choice of $M = 128$ is fairly generous. As we discuss, we remove a substantial fraction of the 128 features initially discovered—and in fact, at any particular time, most of the dataset can be covered by less than 64 features, even when generic features are removed (see Figure 8). However, we are intentionally conservative with this choice of $M$: the set of features that provide 75% coverage in January 2023 is distinct from the set of features that provide 75% coverage in November 2025, which a higher $M$ can accommodate. Moreover, allowing a higher $M$ enables not just the discovery of more granular and nuanced features, but also those that appear only transiently in the dataset (such as short-term spikes) that might otherwise be absorbed into other features for smaller $M$.

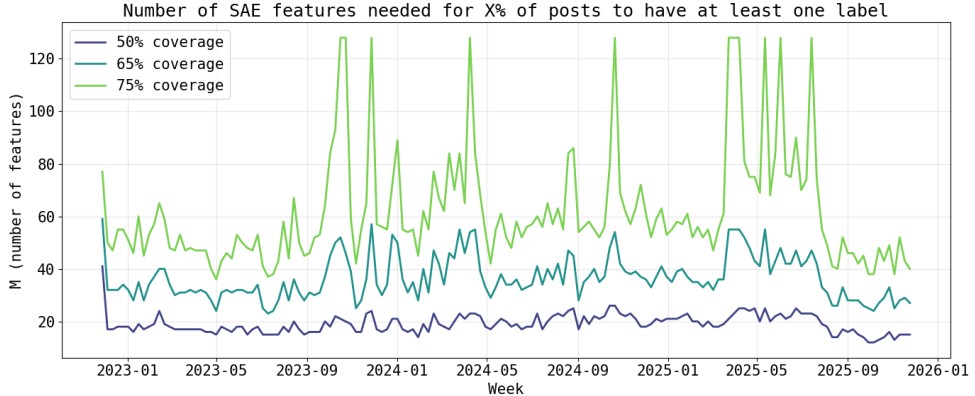

*Figure 8. Number of features needed to "cover" most data in a given week, over time.*

## B.2. Details for temporal trajectories

We analyze trajectories according to the procedure below.[20] Within this section, we use $\{y_t\}_{t\in[T]} :=$ $\frac{1}{|X_t|}\sum_{X\in X_t}\mathbf{1}_{[X \text{ labeled with } i]}$ to denote the actual transcript of (smoothed) daily frequencies, and describe the process for a single feature (dropping the feature index $i$ for clarity).

### B.2.1. FITTING CHANGEPOINTS

To characterize temporal dynamics, we fit piecewise-linear trends to each feature's daily label frequency smoothed (with 30-day rolling mean) according to Equation (1). That is, we approximate $\{y_t\}_{t\in[T]}$ with

$$\lambda(t) = \beta_0 + \sum_{j\in|\mathcal{T}|} \gamma_j \max(0, t - \tau_j).$$

---

[20]Since we model all trajectories as solely (piecewise) linear, our approach is less well-suited to describe features that exhibit temporary "spikes", or brief surges of discussion about specific topics; that said, we are mainly interested in long-run changes.

**Fitting a single trajectory.** Given an active set $S \subseteq [|\mathcal{T}|]$ of changepoints, we fit $(\beta_0, \{\gamma_j\}_{j \in S})$ by minimizing the Poisson NLL

$$\mathcal{L}(S) = \min_{\beta_0, \{\gamma_j\}_{j \in S}} \sum_{t=0}^{T} \left[ \lambda(t) - y_t \log \lambda(t) \right] \quad \text{s.t.} \quad \lambda(t) \geq \varepsilon \; \forall t, \tag{2}$$

with $\varepsilon = 10^{-8}$ being a positivity constraint. Equation (2) is convex in $(\beta_0, \gamma_S)$ and is solved with `cvxpy`.

We select $S$ by dynamic programming over $\mathcal{T}$: for each $s = 0, \ldots, k_{\max}$ with $k_{\max} = 12$, the DP returns the size-$s$ active set $\widehat{S}_s$ that minimizes $\mathcal{L}(S)$ in (2). We iterate upward starting with $s = 0$ and stop at the smallest $s$ for which the relative improvement falls below $\eta = 0.01$, i.e., $s^\star = \min \left\{ s : \frac{\mathcal{L}(\widehat{S}_s) - \mathcal{L}(\widehat{S}_{s+1})}{\sum_t y_t} < \eta \right\}$.

After initial selection, we prune changepoints that appear to have minimal impact on "slope." For each $j \in \widehat{S}_{s^\star}$, define $V_j = \frac{|\widehat{\gamma}_j| \cdot \min(\Delta_{\text{prev}}, \Delta_{\text{next}})}{r_y}$, where $\Delta_{\text{prev}}$ (resp., $\Delta_{\text{next}}$) is the number of days between $\tau_j$ and the *previous* (resp., *next*) neighbor of $\tau_j$ in $\widehat{S}_{s^\star}$, and $r_y = y_{\max} - y_{\min}$ is the full $y$-range. We drop any $j$ where $V_j \leq 0.05$, i.e. where the change in the fitted values before or after $\tau_j$ is less than 5% of the total $y$-range.

**Bootstrapping.** For $b \in [100]$, we draw $N$ post indices with replacement from the $N$ posts, recompute $\{y_t^{(b)}\}_t$, and run the procedure above to obtain $\widehat{S}^{(b)} \subseteq \mathcal{T}$. For each $\tau_j \in \mathcal{T}$, $\text{stab}(\tau_j) = \frac{1}{100} \sum_{b=1}^{100} \mathbf{1}\left[ j \in \widehat{S}^{(b)} \right] \in [0, 1]$, and we use a threshold $\text{stab}(\tau_j) \geq 0.5$ as a heuristic to consider $\tau_j$ *stable*.

### B.2.2. FULL-RANGE SLOPE TESTS

For each feature $i$, we test whether the early-to-late relative change in $y_t^i$ exceeds $\theta = 1.10$. Considering the full time range as a single sequence, we fit $y_t = a + b\,t + \varepsilon_t$ by OLS with Newey-West HAC standard errors at bandwidth $L = \max\left( \lfloor 4(T/100)^{2/9} \rfloor, w_{\text{smooth}} \right)$, where $w_{\text{smooth}} = 30$ covers the rolling-mean autocorrelation. Let $\widehat{r} = (\widehat{a} + \widehat{b} \cdot T)/\widehat{a}$ be the fitted relative change. We run a one-sided $t$-test of

$$\mathcal{H}_0 : \tfrac{1}{\theta} \leq \widehat{r} \leq \theta \quad \text{vs.} \quad \mathcal{H}_1 : \widehat{r} > \theta \text{ or } \widehat{r} < 1/\theta,$$

choosing the side from $\text{sign}(\widehat{b})$. Reported $p$-values are Bonferroni-corrected across 86 features at $\alpha = 0.05$.

Some features are non-monotonic but have "peaks" early (in 2023) or late (in 2025) in the timeframe. To fairly compute the slope tests in these cases, we check for changepoints that have positive slope before, negative slope after, and are either in 2023 or in 2025. For such features, we use these changepoints as the start/end of the slope test instead.

### B.3. Details for finding families of related features

Similarity in *co-occurrence* is measured as the Pearson correlation between label vectors across all posts. $S_{ij}^{\text{corr}} = \text{corr}(a_i, a_j)$ where $a_i = (a_i^{(1)}, \ldots, a_i^{(T)})$ is the vector of feature $i$'s label across all $T$ posts. For similarity in *trajectory*, feature trajectories were aggregated into weekly means and z-score normalized across time. Pairwise trajectory similarity was then computed as the Pearson correlation between these normalized time series, i.e., $S_{ij}^{\text{traj}} = \text{corr}(\widetilde{z}_i, \widetilde{z}_j)$, where $\widetilde{z}_i = \frac{\overline{a}_i^{(t)} - \mu_i}{\sigma_i}$ is the z-scored weekly mean label of feature $i$, and $\overline{a}_i^{(t)} = \frac{1}{|W_t|} \sum_{d \in W_t} a_i^{(d)}$ is the mean label over week $t$.

We use scipy's built-in hierarchical clustering implementation (with the Ward method) to compute clusterings using $S_{ij}^{\text{corr}}$ alone, $S_{ij}^{\text{traj}}$, and an equally-weighted combination of the two (i.e., $S_{ij}^{\text{combined}} = \frac{1}{2}(S_{ij}^{\text{corr}} + S_{ij}^{\text{traj}})$). We compare these clustering approaches on the *emotion* subset of features in Appendix C.2, and compare them to our overall manual categorizations in Appendix F.

### B.4. Matching features across different featurizations

Learned featurizations are unordered—that is, for two different featurizations $C_A$ and $C_B$, a feature indexed as $i$ in $C_A$ has no inherent relationship to (e.g.) $C_B^{(i)}$. Thus, we must manually match features between different featurizations in order to measure the degree to which they identify similar features. Intuitively, two representations of the same feature should have similar activation profiles; features about the concept "login problems" should activate strongly on posts about login problems and not at all on posts that are not. Thus, given two activation matrices $C_A(X)$ and $C_B(X)$ on the same set of data $X$, we would like to compute a matching between the indices in $A$ and those in $B$.

**Step 1: Computing similarities between $A$ and $B$.** Let $C_A, C_B \in \mathbb{R}^{m \times n}$ denote activation matrices from two learners evaluated on the same $n$ posts. We define a similarity matrix $S \in \mathbb{R}^{m \times m}$ using cosine similarity across posts, where index $j$ refers to a feature from $C_A$ and index $k$ refers to a feature from $C_B$ as

$$S_{j,k} = \frac{\langle C_A^{(j)}, C_B^{(k)} \rangle}{\|C_A^{(j)}\| \, \|C_B^{(k)}\|}.$$

**Step 2: Constructing a null distribution of matched similarities.** To quantify the degree to which a feature matching improves upon the baseline of completely random matchings, we run a permutation test as follows. For each of $P$ permutations, we randomly shuffle the activations of each feature in $C_B$ independently across posts, compute the resulting similarity matrix $S^{(\text{null})}$, and extract optimal matching assignments via the Hungarian algorithm.

We choose $\tau$ to be the $(1-\alpha)$ quantile of all matched null similarities over the $P$ permutations; a pair $(j, k)$ matching a feature from $A$ to a feature from $B$ is considered a significant match only if $S_{j,k} \geq \tau$.

**Step 3: Characterizing matches.** Rather than forcing a one-to-one matching upfront, we construct a bipartite graph $G = (V_A \cup V_B, E)$, where $V_A$ indexes features from $C_A$, $V_B$ indexes features from $C_B$, and an edge $(j, k) \in E$ exists whenever $S_{j,k} \geq \tau$.

We then decompose $G$ into connected components $G^{(c)}$ so that $G = \cup_c G^{(c)}$. For a connected component $G^{(c)}$, let $V_A^{(c)} \subseteq V_A$ and $V_B^{(c)} \subseteq V_B$ denote the subsets of vertices from the two sides that appear in that component. We then characterize each component by the sizes of $V_A^{(c)}$ and $V_B^{(c)}$.

- **Match**: $|V_A^{(c)}| = 1, |V_B^{(c)}| = 1$: indicates feature in $A$ is similar to one feature in $B$

- **Split**: $|V_A^{(c)}| = 1, |V_B^{(c)}| > 1$: indicates feature in $A$ is similar to multiple features in $B$

- **Merge**: $|V_A^{(c)}| > 1, |V_B^{(c)}| = 1$: indicates feature in $B$ is similar to multiple features in $A$

- **Unmatched**: $|V_A^{(c)}| = 1, |V_B^{(c)}| = 0$: indicates features in $A$ that are not present in $B$, and vice versa.

Finally, when $|V_A^{(c)}| > 1, |V_B^{(c)}| > 1$, this indicates a **many-to-many** matching. In these cases, we apply the Hungarian algorithm to this subgraph to extract a primary 1-to-1 matching, and classify remaining edges as splits or merges.

**Step 4: Pruning.** We prune matched edges from the previous step as follows. A split or merge edge $(j, k)$ is discarded if both endpoints have strictly higher-scoring alternatives elsewhere in the graph.

### B.5. SAEs vs. other featurization methods

Section 2 defines a featurization as a map $C : [0, 1]^d \to [0, 1]^m$, and the analysis in Section 3 uses sparse autoencoders (SAEs) to instantiate that featurization; however, in principle, all parts of our analysis could have been done with different methods for $C$. Here, we briefly show how alternative methods for computing the featurization—k-means and PCA—can be used to compute activation transcripts $\{C^{(i)}(X)\}_{i \in [m]}$, and validate that they find similar sets of features to those presented in Section 3.

**k-means.** Let $E \in \mathbb{R}^{n \times d}$ denote the embedding matrix. The k-means algorithm partitions the embedding space into $m$ clusters by minimizing within-cluster squared Euclidean distance, producing centroids $\{c_j\}_{j=1}^m$. For each post embedding $e_i$, we define its activation as negative squared distance to each centroid $j$ as $C_{\text{kmeans}}^{(j)}(X_i) = -\|e_i - c_j\|_2^2$. This produces an activation matrix $C_{\text{kmeans}}(X) \in \mathbb{R}^{m \times n}$.

**PCA.** PCA identifies the orthogonal linear basis that captures maximal variance in the given embeddings; this means that individual principal components can often represent distinct concepts in each of its directions ("positive" and "negative"). Thus, to arrive at $m$ features, we run PCA to find $m/2$ principal components and treat each direction of each PC as a separate feature. Let $\{u_k\}_{k=1}^{m/2}$ denote the principal components; then, each principal component $k \in [m/2]$ is split into an activation transcript for its corresponding positive and negative directions as $C_{\text{pca}}^{(2k)}(X_i) = \max(0, -p_{k,i})$ and $C_{\text{pca}}^{(2k+1)}(X_i) = \max(0, p_{k,i})$, where $p_{k,i} = \langle u_k, e_i - \mu \rangle$ and $\mu$ denotes the empirical mean embedding.

**Empirical results.** In all experiments, the embedding model and dataset are identical to those used in Section 2. The feature dimension is fixed to $m = 128$ for comparability. We do not retune hyperparameters for the alternative learners and instead use their standard configurations, fixing only the number of features. In Table 1, we show results from the procedure described in Section B.4 to PCA and k-means.

| Alt. alg | 1-1 matches | SAE features (splits/merges) | Alt. features (splits/merges) | SAE-only | alt-only |
|---|---|---|---|---|---|
| *k-means* | 83 | 39 | 35 | 6 | 10 |
| *PCA* | 76 | 42 | 44 | 10 | 8 |

*Table 1. Comparison of SAE features with alternative algorithms.*

Features that were important to our findings in this paper, e.g. "therapy" or various applications, are consistent across all three methods for featurization. Taken together, these results suggest that the analysis in the main body of this paper is not SAE-specific, but instead reflects stable structure present in the dataset. The small number of unmatched features is consistent with the inherent randomness in unsupervised optimization and does not materially affect the overall feature structure recovered across methods.

## C. Supporting materials for main results

### C.1. Section 3.1 ("domestication")

In Tables 2 and 3, we provide our full categorization of adoption- and (non-emotional) usage- related features. Plots of these features are also provided in Appendix F.

**Basic use** features, which suggest initial exploration of ChatGPT as new users begin using the product, are generally constant or decline over time. **Advanced usage** features, which suggest more involved usage, increase in frequency. Features categorized in **Customization** and **Model or product improvements** both decline, both perhaps due to a combination of increased expertise and improved models. **Temporary bugs** appear fairly consistently. **Applications** decrease overall, with the exception of *medical conditions and diagnoses*, which increases.

Adoption-related features have less-consistent trends within categories. **Language and terminology** features that suggest specific word choice generally decline or remain constant. Explicit references to the **Subreddit community** appear to decline, though the only feature that passes the significance threshold is about sharing user projects, which is likely due in part to a known subreddit policy change in mid-2024. **Perspectives** include users' judgments on the ChatGPT product and/or its societal context that are not directly tied to specific instances of usage. **Product updates** are direct (negative) responses to recent product updates, while **Jailbreaking & content policy** features often peak either at the beginning of the timeline (in early 2023) or near the end (around the GPT-5 release).

These categorizations are our qualitative interpretation of our quantitative results. If features appeared to reasonably belong to multiple categories based on both meaning and quantitative results (e.g., "hallucinations" in both advanced usage and in language and terminology, or "medical applications" in both applications and emotional engagement), we deferred to our judgment of the feature's meaning by reading sample posts. We acknowledge that there may be alternative categorizations, but believe they would not substantively affect our overall interpretation.

### C.2. Section 3.2 (emotional engagement)

**Main emotional engagement features.** Table 4 provides quantitative results for main emotional engagement features. Table 5 lists all features that at least one of our quantitative methods groups with the emotional engagement family.

| Feature | Early | Late | Change | Changepoint |
|---|---|---|---|---|
| *poetic language*[†] | 1.0% | 6.8% (Apr'25) | ↑ × 3.2 | ⌣ 2024-07-30 |
| *feelings of attachment or companionship with AI* | 0.6% | 3.8% (Apr'25) | ↑ × 4.8 | ⌣ 2024-05-13 |
| *using ChatGPT for emotional support or therapy* | 0.7% | 3.0% | ↑ × 5.1 | ⌣ 2024-05-13 |
| *naming ChatGPT* | 0.5% | 1.4% (Apr'25) | ↑ × 2.3 | |
| *romantic relationships with AI* | 0.4% | 0.9% | ↑ × 3.0 | |
| *AI consciousness or sentience* | 1.5% | 2.8% (Apr'25) | × 1.2 | ⌣ 2024-07-30 |
| *personal stories about positive impact* | 2.8% | 3.1% (Jan'25) | × 0.86 | ⌣ 2024-05-13 |

Table 4. *Emotional engagement features. Early: Jan 2023. Late: Nov 2025. For features that peak before the end of 2025, month of peak is noted. Gray rows have $p > 0.05$, for test of slope change $\geq 10\%$; features marked with † have $p < 0.05$ but $p_{adj} \geq 0.05$ after Bonferroni correction.*

| Feature | Category (see Tables 2, 3) | traj. | corr. | comb. | chpt. |
|---|---|---|---|---|---|
| *medical conditions* | applications | ✓ | ✓ | ✓ | ✓ |
| *requests for harsh or unfiltered roasts* | uncategorized | ✓ | | ✓ | |
| *memory features and data saving* | advanced usage | ✓ | | | ✓ |
| *polite expressions ("please" and "thank you")* | language | ✓ | | | |
| *offensive or inappropriate content* | jailbreaking | ✓ | | | |
| *false or fabricated information* | advanced usage | ✓ | | | |
| *societal collapse and existential-threat scenarios* | perspectives | ✓ | | | |

Table 5. *"emotional engagement" features identified by automated methods only.*

**Therapy vs companionship.** Table 6 shows the top 20 most distinctive words for *therapy* versus *companionship*.

Table 6. *Distinctive words for* therapy *feature vs* companionship *feature. Log-odds computed with informative Dirichlet priors. $n_{therapy}$ and $n_{companion}$ show raw counts in therapy-only ($n = 1889$) vs companionship-only ($n = 2561$) posts.*

*(a) More distinctive of therapy*

| term | log-odds | $z$-score | $n_{therapy}$ | $n_{companion}$ |
|---|---|---|---|---|
| therapist | −2.26 | −24.4 | 1319 | 59 |
| therapy | −2.47 | −22.8 | 1166 | 31 |
| help | −0.98 | −19.0 | 1406 | 403 |
| mental | −1.99 | −18.8 | 792 | 58 |
| health | −2.22 | −18.1 | 723 | 35 |
| helped | −1.65 | −16.2 | 629 | 77 |
| my | −0.31 | −15.6 | 5919 | 3664 |
| for | −0.31 | −15.2 | 5498 | 3388 |
| life | −0.80 | −15.2 | 1181 | 418 |
| support | −1.26 | −14.3 | 604 | 123 |
| her | −0.67 | −11.2 | 835 | 343 |
| through | −0.68 | −11.2 | 824 | 337 |
| trauma | −1.99 | −11.1 | 274 | 20 |
| anxiety | −2.09 | −10.8 | 260 | 16 |
| issues | −1.31 | −10.6 | 318 | 61 |
| advice | −0.96 | −10.5 | 443 | 130 |
| she | −0.59 | −10.5 | 897 | 407 |
| was | −0.26 | −10.0 | 3260 | 2124 |
| years | −0.99 | −9.9 | 375 | 106 |
| professional | −1.37 | −9.8 | 265 | 47 |

*(b) More distinctive of companionship*

| term | log-odds | $z$-score | $n_{therapy}$ | $n_{companion}$ |
|---|---|---|---|---|
| explicitly | 2.89 | 37.7 | 24 | 2709 |
| like | 0.61 | 26.2 | 2541 | 4396 |
| personality | 1.47 | 17.4 | 133 | 632 |
| it | 0.19 | 15.8 | 12522 | 13443 |
| feels | 0.95 | 14.6 | 280 | 713 |
| just | 0.37 | 13.9 | 2212 | 2918 |
| human | 0.64 | 13.8 | 649 | 1154 |
| conversation | 0.72 | 11.9 | 369 | 718 |
| humans | 1.06 | 11.6 | 139 | 400 |
| feel | 0.39 | 11.0 | 1254 | 1683 |
| bing | 2.08 | 10.7 | 18 | 202 |
| gpt | 0.37 | 10.5 | 1226 | 1623 |
| else | 0.63 | 10.4 | 385 | 675 |
| something | 0.41 | 10.1 | 929 | 1284 |
| question | 0.83 | 9.8 | 179 | 394 |
| friend | 0.56 | 9.7 | 438 | 710 |
| more | 0.32 | 9.6 | 1448 | 1807 |
| its | 0.49 | 9.5 | 564 | 847 |
| tone | 0.85 | 9.3 | 150 | 341 |
| anyone | 0.51 | 9.0 | 458 | 706 |

| Feature | Overall rate | *therapy* rate | *companion* rate | $\frac{therapy}{companion}$ ratio |
|---|---|---|---|---|
| *positive impact* | 1.8% | 20% (× 11.6) | 4.9% (× 2.8) | 4.2 (3.5, 5.1) |
| *privacy concerns* | 1.6% | 3.5% (× 2.2) | 0.4% (× 0.3) | 8.3 (4.4, 15.6) |
| *naming ChatGPT* | 0.8% | 0.4% (× 0.5) | 3.6% (× 4.5) | 0.1 (0.05, 0.2) |
| *AI sentience* | 1.8% | 0.8% (× 0.4) | 6.2% (× 3.5) | 0.1 (0.08, 0.2) |
| *recent quality decline* | 3.0% | 1.0% (× 0.3) | 6.6% (× 2.2) | 0.2 (0.09, 0.2) |

*Table 7. How frequently* therapy-*only and* companion-*only posts also exhibit other features (rows). Rate shows overall prevalence; lifts (×) show how much more frequently each column feature co-occurs with each row feature, compared to all posts. "Ratio" column compares therapy ÷ companion; 95% CIs for ratio, modeling counts as Bernoulli trials, shown in parentheses.*

| Category | Feature | Early | Late | Change |
|---|---|---|---|---|
| Basic use and exploration | recommendations for AI tools | 5.8% (Jul'23) | 2.2% | ↓ × 0.28 |
| | questions about access, versions, pricing | 5.3% | 2.5% | ↓ × 0.29 |
| | login problems | 2.0% | 1.6% | ↓ × 0.37 |
| | requests for help | 3.7% | 2.8% | ↓ × 0.67 |
| | *questions about trying specific features* | *2.0%* | *1.6%* | *× 0.71* |
| | *model or version preference comparisons* | *0.7%* | *1.7%* | *× 0.71* |
| | *pricing and free vs paid comparisons* | *4.0%* | *5.5%* | *× 0.91* |
| Advanced usage | false or fabricated information | 1.7% | 2.8% | ↑ × 1.3 |
| | organizing or searching chat histories | 1.5% | 3.2% | ↑ × 1.6 |
| | requests to turn specific features off | 1.8% | 4.0% | ↑ × 1.7 |
| | lost, deleted, or missing conversations | 1.7% | 3.2% | ↑ × 1.9 |
| | memory features and data saving | 0.8% | 3.1% | ↑ × 3.8 |
| | cross-chat data leaks | 0.3% | 3.1% | ↑ × 5.8 |
| | hallucinations | 0.2% | 1.4% | ↑ × 8.0 |
| | failing to follow user instructions[†] | 1.5% | 4.2% | ↑ × 1.6 |
| | *AI recognizing or admitting mistakes* | *2.7%* | *2.5%* | *× 0.68* |
| | *formatting and copy-paste issues* | *0.2% (May'23)* | *0.1%* | *× 0.82* |
| | *tool usage questions* | *1.1%* | *1.6%* | *× 1.1* |
| | *questions about daily or repeated AI use* | *2.5%* | *2.3%* | *× 1.1* |
| Customization | tools and extensions | 2.1% | 0.2% | ↓ × 0.00 |
| | fine-tuning GPTs with user-provided data | 0.9% | 0.1% | ↓ × 0.08 |
| | prompts and prompting | 6.4% | 3.3% | ↓ × 0.38 |
| | custom instructions[†] | 2.4% (Sep'23) | 0.7% | ↓ × 0.10 |
| Model or product improvements | knowledge cutoff discussions | 0.9% | 0.6% | ↓ × 0.23 |
| | browser issues or browser extensions | 1.5% | 1.3% | ↓ × 0.35 |
| | message limits or caps | 3.3% | 1.5% | ↓ × 0.54 |
| | *PDF upload or summarization* | *0.4%* | *0.8%* | *× 0.65* |
| Temporary bugs | slow or lagging response times | 1.1% | 2.2% | ↑ × 1.7 |
| | error messages and technical problems | 3.0% | 7.2% | ↑ × 1.8 |
| | *ChatGPT down or unavailable* | *3.3%* | *2.0%* | *× 0.35* |
| | *ChatGPT failing to process inputs* | *3.2%* | *3.9%* | *× 0.93* |
| Applications | programming | 5.0% | 1.6% | ↓ × 0.18 |
| | education or studying | 5.1% | 1.2% | ↓ × 0.19 |
| | AI text detection for student work | 1.4% | 0.5% | ↓ × 0.20 |
| | D&D and role-playing games | 1.2% | 0.4% | ↓ × 0.24 |
| | math and problem-solving | 1.9% | 0.8% | ↓ × 0.29 |
| | songwriting | 1.9% | 0.8% | ↓ × 0.36 |
| | riddles and logic problems | 1.1% | 0.3% | ↓ × 0.40 |
| | job applications and resumes | 0.6% | 0.4% | ↓ × 0.42 |
| | marketing, advertising, business growth | 2.4% | 1.5% | ↓ × 0.51 |
| | language use, translation, multilingual | 1.4% | 1.4% | ↓ × 0.53 |
| | medical conditions or diagnoses[‡] | 0.4% | 1.4% | ↑ × 2.4 |
| | investing, finance, or wealth topics[†] | 1.2% | 0.8% | ↓ × 0.59 |
| | movies, posters, and film[†] | 1.5% | 0.4% | ↓ × 0.61 |
| | *creative writing* | *6.0%* | *2.5%* | *× 0.31* |
| | *legal advice and lawsuits* | *1.0% (Jul'23)* | *0.7%* | *× 0.40* |
| | *religion or religious texts* | *0.7%* | *0.5%* | *× 0.86* |
| | *maps or geographic information* | *1.2%* | *1.8%* | *× 0.97* |

*Table 2. Frequency of posts exhibiting each feature, by category. Early: mean monthly percentage in Jan 2023. Late: Nov 2025. For features that peak then decline, month of peak is noted and used instead. Gray rows have $p > 0.05$ for slope test (not significant even before correction). Features marked with † have $p < 0.05$ but $p_{adj} \geq 0.05$ after Bonferroni correction ($\times 86$). All remaining features are significant after Bonferroni correction. Features marked with ‡ show opposite trends relative to their category. "Change" column is the relative change suggested by the fitted model.*

| Category | Feature | Early | Late | Change |
|---|---|---|---|---|
| Language and terminology | *mentions google (search)* | 2.0% | 0.6% | ↓ × 0.17 |
| | *uses the word "bot" or "chatbot"* | 3.0% | 1.3% | ↓ × 0.29 |
| | *use of the word "generate" or variants*[†] | 2.0% | 1.1% | ↓ × 0.62 |
| | *polite expressions ("please" and "thank you")* | 0.8% | 0.4% | × 0.55 |
| | *uses the word "dumb" or similar* | 1.1% | 1.0% | × 0.92 |
| Subreddit community | *user-built projects (sharing or feedback)* | 4.2% | 1.7% | ↓ × 0.20 |
| | *feature suggestions or improvement requests* | 2.7% | 1.1% | × 0.35 |
| | *reference to Reddit explicitly* | 2.2% | 1.3% | × 0.51 |
| | *"why" questions about others' attitudes* | 1.3% | 1.2% | × 0.82 |
| Perspectives | *discussions about how LLMs represent knowledge* | 3.3% | 1.3% | ↓ × 0.20 |
| | *predictions about future development or capabilities* | 5.6% | 1.7% | ↓ × 0.28 |
| | *ethical, legal, or copyright concerns* | 2.3% | 0.7% | ↓ × 0.38 |
| | *privacy concerns (data leaks or exposure)* | 0.8% | 2.6% | ↑ × 1.7 |
| | *perceived bias in ChatGPT responses*[†] | 1.6% | 0.5% | ↓ × 0.33 |
| | *societal collapse and existential-threat scenarios*[†] | 1.9% | 1.3% (Apr'25) | ↓ × 0.91 |
| | *societal impacts, risks, controversies* | 4.5% | 4.4% | × 0.85 |
| Product updates | *perception of recent drops in quality*[†] | 3.9% (Nov'23) | 5.0% | ↑ × 1.0 |
| | *frustration or hatred about product updates*[†] | 2.4% | 7.8% | ↑ × 2.5 |
| | *dissatisfaction with 4o removal and loss of control*[†] | 0.3% | 7.6% (Aug'25) | ↑ × 4.5 |
| Jailbreaking & content policy | *jailbreak prompts or techniques* | 3.8% (Mar'23) | 0.4% | ↓ × 0.00 |
| | *offensive or inappropriate content* | 0.1% | 0.5% | ↑ × 3.3 |
| | *jailbreaking via DAN or personas* | 2.0% (Mar'23) | 0.3% | × 0.09 |
| | *censorship or content policy restrictions* | 5.9% | 5.4% | × 0.59 |
| | *NSFW content* | 1.6% | 1.9% | × 0.82 |
| | *complaints about getting direct or unfiltered answers* | 1.7% | 2.8% | × 1.2 |

*Table 3. Adoption-related features. See Table 2 for column definitions and significance notation.*

# D. Algorithms and proofs for Section 4

The core algorithmic tool that PULSE relies on is a "betting-style" algorithm for sequential mean testing. One such algorithm is the one implemented in Dai et al. (2025a), which is summarized in Algorithm 2.

---

**Algorithm 2** *Level-$\alpha$ sequential mean test for $\mathcal{H}_0 : \mu \leq \mu_0$*

**Procedure** INITIALIZE($\mu_0$, $\alpha$)
|   $\omega \leftarrow 0; \quad \lambda \leftarrow 0; \quad S \leftarrow 0$
**Procedure** INCREMENT($x$)
|   $z \leftarrow \frac{x - \mu_0}{1 + \lambda(x - \mu_0)}$
|   $\omega \leftarrow \omega + \ln(1 + \lambda(x - \mu_0))$
|   $S \leftarrow S + z^2$
|   $\lambda \leftarrow \text{Proj}_{[0,1]}\left(\lambda + \frac{2}{2 - \ln(3)} \cdot \frac{z}{1 + S}\right)$
|   **return** $\omega > \ln(1/\alpha)$                           `// reject` $\mathcal{H}_0$

---

Algorithm 2 generically provides the following guarantee. This result is elementary (see, e.g., proof of a very similar result in Dai et al. (2025a)); we reproduce it here in the context of our paper for completeness.

**Theorem D.1** (Validity). *Let $x_1, x_2, \ldots \in [0, 1]$ be a stream of observations with $\mathbb{E}[x_t \mid \mathcal{F}_{t-1}] \leq \mu$, where $\mathcal{F}_{t-1}$ is the filtration generated by $x_1, \ldots, x_{t-1}$. Running Algorithm 2 at level $\alpha$ on this stream guarantees that, if $\mathcal{H}_0 : \mu \leq \mu_0$ holds, the likelihood of ever rejecting is at most $\alpha$. That is, $\Pr\left[\exists t : \omega_t > \ln(1/\alpha) \text{ when } \mu \leq \mu_0\right] \leq \alpha$.*

*Proof.* First note that when $\mathcal{H}_0$ holds, the sequence $\{\exp(\omega_t)\}_{t \geq 0}$ is a non-negative supermartingale. Non-negativity follows directly from the exponential. The supermartingale property follows from

$$
\begin{aligned}
\mathbb{E}[\exp(\omega_t)|\mathcal{F}_{t-1}] &= \mathbb{E}[\exp(\omega_{t-1} + \ln(1 + \lambda_{t-1}(x_t - \mu_0)))|\mathcal{F}_{t-1}] \\
&= \exp(\omega_{t-1}) \cdot (1 + \lambda_{t-1}(\mathbb{E}[x_t \mid \mathcal{F}_{t-1}] - \mu_0)) \\
&\leq \exp(\omega_{t-1}) \cdot (1 + \lambda_{t-1}(\mu - \mu_0)) \\
&\leq \exp(\omega_{t-1}),
\end{aligned}
$$

where the second equality uses that $\lambda_{t-1}$ is $\mathcal{F}_{t-1}$-measurable (predictable), and the inequality holds because $\mu \leq \mu_0$ under $\mathcal{H}_0$ and $\lambda_{t-1} \geq 0$. Applying Ville's inequality to the supermartingale $\{\exp(\omega_t)\}_{t \geq 0}$ yields

$$
\Pr[\exists t : \omega_t > \ln(1/\alpha)] = \Pr[\exists t : \exp(\omega_t) > 1/\alpha] \leq \mathbb{E}[\exp(\omega_0)] \cdot \alpha = \alpha,
$$

where the final equality follows because $\omega_0 = 0$ and hence $\exp(\omega_0) = 1$. $\qquad\square$

With this in hand, we can give concrete algorithms and guarantees for each of our procedures. We first handle the accuracy problem and the feature tracking problem one at a time, as our statistical guarantees are made separately, and then summarize how to put them together in Section D.4.

## D.1. Accuracy monitoring.

Our concrete procedure for accuracy monitoring is given in Algorithm 3.

---

**Algorithm 3** *Real-time accuracy test*

---

**Input:** Initial data $X_{\text{init}}$, featurization algorithm $\mathcal{A}$, threshold factor $\beta$, significance $\alpha$
**Output:** Sequence of featurizations with timestamps $\{(\widehat{C}_s, t_s)\}_{s \geq 0}$

11    **Initialize:** $s \leftarrow 0$;    $\widehat{C}_{\text{curr}} \leftarrow \mathcal{A}(X_{\text{init}})$;    $\varepsilon_{\text{curr}} \leftarrow \text{err}(\widehat{C}_{\text{curr}}(X_{\text{init}}))$; emit $(\widehat{C}_{\text{curr}}, 0)$
     Let $\tau$ be an instance of Algorithm 2, and call $\tau.\text{INITIALIZE}(\beta \cdot \varepsilon_{\text{curr}}, \alpha/10.58)$.

12    **for** *each new data batch* $X_t$ **do**

13     |    $\varepsilon_t \leftarrow \text{err}(\widehat{C}_{\text{curr}}(X_t))$
         *rejected* $\leftarrow \tau.\text{INCREMENT}(\varepsilon_t)$
         **if** *rejected* **then**

14         |    $s \leftarrow s + 1$, and emit $(\widehat{C}_{\text{curr}}, t)$
             $\widehat{C}_{\text{curr}} \leftarrow \mathcal{A}(X_{1:t})$
             and $\varepsilon_{\text{curr}} \leftarrow \text{err}(\widehat{C}_{\text{curr}}(X_{1:t}))$
             Compute $\alpha_s = \alpha \cdot (s+1)^{-0.1}/10.58$, and set up $\tau.\text{INITIALIZE}(\beta \cdot \varepsilon_{\text{curr}}, \alpha_s)$

15    **return** all emitted $(\widehat{C}_s, t_s)$ pairs

---

For this approach, recall that we test null hypothesis $\mathcal{H}_0^{\text{acc}} : \text{err}(\widehat{C}_{\text{curr}}(X_t)) \leq \beta \cdot \varepsilon_{curr}$. The key modeling assumption in order to apply Theorem D.1 is that the sequence of errors $\varepsilon_t$ satisfies $\mathbb{E}[\varepsilon_t \mid \mathcal{F}_{t-1}] \leq \beta \varepsilon_{\text{curr}}$.[21] The following proposition formalizes the FDR guarantee.

**Proposition D.2** (Formal statement of Proposition 4.1)**.** *Run Algorithm 3 with the $s$-th test (using Algorithm 2) at level* $\alpha_s = \alpha \cdot \frac{(s+1)^{-0.1}}{10.58}$, *as specified in line 10. Let $\mathcal{R}_S$ be the set of test indices that reject among the first $S$ tests, and let* $\mathcal{H}_0 \subseteq \mathbb{N}$ *denote the set of indices where $\mathcal{H}_0^{\text{acc}}$ holds. Then,* $\text{FDR}(\mathcal{R}_S) := \mathbb{E}\left[\frac{|\mathcal{R}_S \cap \mathcal{H}_0|}{|\mathcal{R}_S| \vee 1}\right] \leq \alpha$ *for all $S \in \mathbb{N}$.*

The proof of Proposition D.2 adapts Theorem 1 of Xu & Ramdas (2024), restated here, to our setting.

**Theorem D.3** (Theorem 1 of Xu & Ramdas (2024), simplified)**.** *Let $(E_s)_{s \in \mathbb{N}}$ be a sequence of e-values satisfying $\mathbb{E}[E_s] \leq 1$ for each $s \in \mathcal{H}_0$, where $\mathcal{H}_0$ denotes the set of true null hypotheses. Let $(\gamma_s)_{s \in \mathbb{N}}$ be a non-negative sequence with $\sum_{s=0}^{\infty} \gamma_s \leq 1$. Define adaptive test levels $\alpha_s := \alpha \gamma_s(|\mathcal{R}_{s-1}| + 1)$, where $\mathcal{R}_{s-1}$ is the set of rejections among tests $0, \ldots, s-1$, and reject test $s$ when $E_s \geq 1/\alpha_s$. Then $\text{FDR}(\mathcal{R}_S) \leq \alpha$ for all $S \in \mathbb{N}$.*

*Proof of Proposition D.2.* By Theorem D.1, for each test $s$ where $\mathcal{H}_0^{\text{acc}}$ holds, the wealth process $\{\exp(\omega_t)\}_{t \geq 0}$ is a non-negative supermartingale with $\exp(\omega_0) = 1$. Let $T_s$ denote the stopping time at which test $s$ rejects (with $T_s = \infty$ if it never rejects), and define the e-value $E_s := \exp(\omega_{T_s})$. By optional stopping, $\mathbb{E}[E_s] \leq 1$ when $\mathcal{H}_0^{\text{acc}}$ holds for test $s$.

A key observation is that we test $s$ only when test $s - 1$ rejects; this makes the sequence of tests defined by our algorithm a special case of Theorem D.3. When test $s$ rejects (i.e., $\mathbf{1}\{E_s \geq 1/\alpha_s\} = 1$), all tests $0, 1, \ldots, s$ have rejected, so $|\mathcal{R}_S| \geq s + 1$. This implies that $\frac{\mathbf{1}\{E_s \geq 1/\alpha_s\}}{|\mathcal{R}_S| \vee 1} \leq \frac{\mathbf{1}\{E_s \geq 1/\alpha_s\}}{s+1}$. Combining these observations, we have

$$\text{FDR}(\mathcal{R}_S) = \sum_{s \in \mathcal{H}_0 \cap [S]} \mathbb{E}\left[\frac{\mathbf{1}\{E_s \geq 1/\alpha_s\}}{|\mathcal{R}_S| \vee 1}\right] \leq \sum_{s \in \mathcal{H}_0 \cap [S]} \mathbb{E}\left[\frac{\alpha_s E_s}{s+1}\right] = \sum_{s \in \mathcal{H}_0 \cap [S]} \frac{\alpha \cdot (s+1)^{-0.1}}{10.58(s+1)} \mathbb{E}[E_s] \leq \frac{\alpha}{10.58} \sum_{s=0}^{\infty} \frac{1}{(s+1)^{1.1}} \leq \alpha,$$

where the first inequality applies the denominator bound, and the last uses $\sum_{s=1}^{\infty} s^{-1.1} = \zeta(1.1) \approx 10.58$. $\qquad\square$

### D.2. Feature monitoring.

For feature monitoring, our algorithm can be formalized as follows.

---

[21] The astute reader may notice that future errors should generally be expected to exceed prior error, simply due to having fit the model to optimize the prior data. While this can statistically be resolved by sample splitting, we feel that the tradeoff, e.g., in reduced model quality, would not justify the slightly improved statistical "rigor." Realistically, $\beta$ can easily be set high enough to exceed the expected additional error due to generalization.

---

**Algorithm 4** *Feature monitoring with dynamic active set*

---

**Procedure** INITIALIZE($\widehat{C}_{curr}, \alpha, \beta, S_{init}$)

  | Set $\widehat{C}_{\mathrm{curr}}, \alpha, \beta$ and call UPDATE($S_{\mathrm{init}}, \emptyset$)

**Procedure** INCREMENT($X_t$)

  | *rejected* = {}

  | **for** *each* $i \in S$ **do**

    | $r \leftarrow \tau^{(i)}$.INCREMENT($\widehat{C}_{\mathrm{curr}}^{(i)}(X_t)$) **if** *r* **then** *rejected* $\leftarrow$ *rejected* $\cup\, i$

  | **return** *rejected*

**Procedure** UPDATE($S_{add}, S_{remove}$)

  | $S \leftarrow S \setminus S_{\mathrm{remove}}$ and $\alpha_{\mathrm{add}} = \sum_{i \in S_{\mathrm{remove}}} \alpha_i$

  | Set $\{\alpha_i\}_{i \in S_{\mathrm{add}}}$ with $\sum_{i \in S_{\mathrm{add}}} \alpha_i \leq \alpha_{\mathrm{add}}$

  | **for** *each* $i \in S_{add}$ **do**

    | Let $\tau^{(i)}$ be an instance of Algorithm 2, and call $\tau^{(i)}$.INITIALIZE($\beta \cdot \widehat{C}_{\mathrm{curr}}^{(i)}(X_{0:t}), \alpha_i$)

---

As with the accuracy monitor, invoking Theorem D.1 also involves some modeling details. Specifically, for feature-specific monitoring, our null hypothesis indicates that $\mathbb{E}\left[\widehat{C}_{\mathrm{curr}}^{(i)}(X_t) \mid \mathcal{F}_{t-1}\right] = \widehat{C}_{\mathrm{curr}}^{(i)}(X_{0:r})$, where $r$ is the time at which the test was initialized and randomness is due to realizations of $X_t$.

The following proposition formalizes the FWER guarantee for Algorithm 4.

**Proposition D.4** (Formal statement of Proposition 4.2)**.** *Initialize Algorithm 4 at level $\alpha$ and run it (i.e., call* INCREMENT *repeatedly) on a stream of observations. Let $S$ be the current active set of features, $r$ be the most recent time at which $S$ was updated, and $\mathcal{R}_t$ be the set of tests rejected by Algorithm 4 at $t$. Then,*

$$\Pr\left[\exists t > r : \exists i \in \mathcal{R}_t \text{ where } \mathbb{E}\left[\widehat{C}_{curr}^{(i)}(X_t)\right] \leq \widehat{C}_{curr}^{(i)}(X_{0:r})\right] \leq \alpha,$$

*even if $S_{add}$, $S_{remove}$, and $r$ are chosen with arbitrary dependence on $\mathcal{F}_r$.*

*Proof.* Fix a run of Algorithm 4 on a sequence of observations, and let $r$ be the most recent time at which the active set $S$ was updated. Let $S(r)$ denote the corresponding active set, and condition on $\mathcal{F}_r$, the filtration containing all randomness until (and including) time $r$.

For each $i \in S(r)$, let $A_i := \left\{\exists t > r : i \in \mathcal{R}_t \text{ and } \mathcal{H}_0^{(i)} \text{ holds}\right\}$. Each instance $\tau^{(i)}$ of Algorithm 2 is by anytime-valid for all samples arriving after $r$ by Theorem D.1; that is, for each null $i \in S(r) \cap \mathcal{H}_0$, we have that $\Pr[A_i \mid \mathcal{F}_r] \leq \alpha_i$. The result follows from union bounding over all $i \in S(r)$, noting that $\sum_{i \in S} \alpha_i \leq \alpha$ by construction, and taking expectations over $\mathcal{F}_r$. $\qquad\square$

*Remark* D.5. The result in Proposition D.4 may, at first glance, appear "too good to be true;" it is well-known that, in general, it is impossible to select hypotheses data-dependently while simultaneously testing them online. For our specific instantiation of the sequential testing framework, however, no samples used for selection are ever re-used for testing, as Line 13 of Algorithm 4 always *resets* each new feature test.

### D.3. Implementation details

**Incorporating release dates.** While monitoring can be done without explicit intervention at pre-specified times, in a realistic monitoring scenario, one might be interested in checking whether changes occur after a model release or update that happens at a known time. In fact, it is fairly straightforward to do so by re-setting the current state of the currently-active hypothesis tests; see Appendix D.

**Comparing featurizations.** As $\widehat{C}_{\mathrm{curr}}$ is updated, one important question is how a new $\widehat{C}_s$ differs from the previous $\widehat{C}_{s-1}$—and, in particular, whether features in $\widehat{C}_s$ might be reasonably "matched" to features $i$ in $\widehat{C}_{s-1}$, or if $\widehat{C}_s$ contains new features not captured in $\widehat{C}_{s-1}$. Since the features learned by typical featurization algorithms (SAEs included) are unordered by default, $\widehat{C}_s^{(i)}$ cannot be compared directly to $\widehat{C}_{s+1}^{(i)}$. Instead, we match features by similarity in activation profile on the same set of data; see Appendix B.4 for details.

### D.4. Combined procedure

Finally, in Algorithm 5, we show how Algorithms 3 and 4 can be used in tandem.

---

**Algorithm 5** *Combined online monitoring (formal version of Algorithm 1)*

---

**Input:** Initial data $X_{\text{init}}$, featurization algorithm $\mathcal{A}$, threshold $\beta$, significance levels $\alpha_{\text{acc}}, \alpha_{\text{feat}}$
**Output:** Sequence of featurizations and feature alerts

16 **Initialize:** $\widehat{C}_{\text{curr}} \leftarrow \mathcal{A}(X_{\text{init}})$; $\varepsilon_{\text{curr}} \leftarrow \text{err}(\widehat{C}_{\text{curr}}(X_{\text{init}}))$
    Let $\tau_{\text{acc}}$ be an instance of Algorithm 2, and call $\tau_{\text{acc}}.\textsc{Initialize}(\beta \cdot \varepsilon_{\text{curr}}, \alpha_{\text{acc}})$
    Let $\mathcal{F}$ be an instance of Algorithm 4, and call $\mathcal{F}.\textsc{Initialize}(\widehat{C}_{\text{curr}}, \alpha_{\text{feat}}, \beta, S_{\text{init}})$ **for** *each new data batch $X_t$* **do**

17    |  $\varepsilon_t \leftarrow \text{err}(\widehat{C}_{\text{curr}}(X_t))$ , and *acc_rejected* $\leftarrow \tau_{\text{acc}}.\textsc{Increment}(\varepsilon_t)$
   |  **if** *acc_rejected* **then**
18    |  |  Alert: accuracy degradation
   |  |  $\widehat{C}_{\text{curr}} \leftarrow \mathcal{A}(X_{1:t})$; $\varepsilon_{\text{curr}} \leftarrow \text{err}(\widehat{C}_{\text{curr}}(X_{1:t}))$
   |  |  New test $\tau_{\text{acc}}.\textsc{Initialize}(\beta \cdot \varepsilon_{\text{curr}}, \alpha_s)$ with adjusted $\alpha_s$
   |  |  Optionally specify $S_{\text{init}}$ and call $\mathcal{F}.\textsc{Initialize}(\widehat{C}_{\text{curr}}, \alpha_{\text{feat}}, \beta_{\text{feat}})$

19    |  **if** *external signal to update $S$ (e.g., model update)* **then**
20    |  |  Specify $S_{\text{new}}, S_{\text{old}}$ and call $\mathcal{F}.\textsc{Update}(S_{\text{new}}, S_{\text{old}})$

21    |  *feat_rejected* $\leftarrow \mathcal{F}.\textsc{Increment}(X_t)$
   |  **if** *feat_rejected* $\neq \emptyset$ **then**
22    |  |  Alert: features *feat_rejected* show significant change

---

# E. Monitoring experiments for Section 4

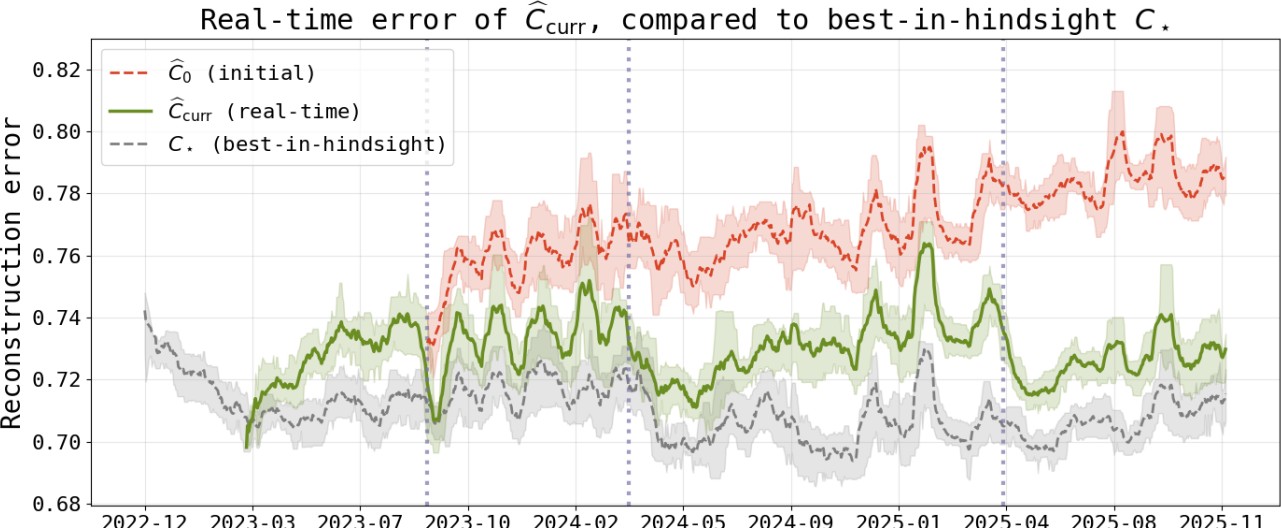

*Figure 9. Reconstruction error of the real-time approach of Section 4.1 ($\widehat{C}_{curr}$), using $\alpha_s$ schedule at $\alpha = 0.1$, compared to reconstruction error of best-in-hindsight $C_\star$ and initial $\widehat{C}_0$. "Reject and retrain" timesteps marked with dotted vertical lines.*

**How did features evolve over time?** As discussed above, each time a new $\widehat{C}_{curr}$ is computed, its features do not automatically correspond to those of the previous featurization.

The evolution of featurizations overall are broadly consistent with known external changes and with the in-hindsight clusterings. For instance, between $\widehat{C}_0$ and $\widehat{C}_1$, two new features emerge corresponding to *plugins* and the *ChatGPT API*, both of which were product updates from March 2023; between $\widehat{C}_1$ and $\widehat{C}_2$, a feature corresponding to *controversy, danger, or bans* disappears, while one for *low-quality AI-generated content* emerges; between $\widehat{C}_2$ and $\widehat{C}_3$, meanwhile, features corresponding to *Google Bard* and *medical topics* disappear, while features corresponding to *Gemini* and *personalized image requests* emerge.

In Figure 10, we show how selected sets of features evolved over $\widehat{C}_0$, $\widehat{C}_1$, $\widehat{C}_2$, and $\widehat{C}_3$.

|  | $\widehat{C}_0 \to \widehat{C}_1$ (2023-09-09) | $\widehat{C}_1 \to \widehat{C}_2$ (2024-04-04) | $\widehat{C}_2 \to \widehat{C}_3$ (2025-04-18) |
|---|---|---|---|
| **Obsolete** | *Reddit poll link included* | *ChatGPT controversy and bans* *translation and language tasks* | *medical topics and terminology* *mentions "my child"* *mentions Bard explicitly* |
| **New** | *free AI tool recommendations* *cooking recipes and meal planning* *unexplained account behavior issues* *ChatGPT plugins access discussions* *API and API key discussions* | *video content help requests* *child creative project mentions* *AI spam and bot content* | *mentions Gemini or Google Gemini* *personalized AI image requests* |

*Table 8. Summary of new and obsolete features at each transition between featurizations $\widehat{C}_s$.*

In Table 9, we give the numeric version of data presented in Figure 4, and also add features related to *sentience* and *spirituality*.

Notably, the quality of feature representations does appear to affect alert times. Using representations $\widehat{C}_0$, *no tests result in alerts*, likely because the representations of the "therapy" feature in $\widehat{C}_0$ are too weak, or otherwise not fully capturing characteristics of later posts about therapy. (No tests result in alerts even for $n = 1$; the bottleneck is the representation, rather than the testing of multiple features simultaneously.)

On the other hand, while alerts for *gratitude towards ChatGPT* (using the $\widehat{C}_1$ representations) would have been raised at similar times to the $\widehat{C}_2$ therapy feature, it is unclear that, at the time, *gratitude* would have been considered a societally-

| Reps. | Test start | Event | Feature | $n = 1$ | $n = 3$ | $n = 5$ | $n = 10$ | $n = 64$ |
|---|---|---|---|---|---|---|---|---|
| $\widehat{C}_1$ | 23-09-23 | $\widehat{C}_1$ computed | *gratitude toward ChatGPT* | 24-10-25 | 24-11-12 | 24-11-20 | 24-11-29 | 25-01-05 |
| | | | *medical and psychological advice* | 24-12-19 | 25-02-19 | 25-03-18 | 25-04-21 | 25-05-30 |
| | | | *AI consciousness and sentience* | 25-02-01 | 25-02-27 | 25-03-12 | 25-03-25 | 25-05-07 |
| | 24-03-04 | Voice chat on apps | *gratitude toward ChatGPT* | 24-10-26 | 24-11-13 | 24-11-21 | 24-11-30 | 25-01-07 |
| | | | *medical and psychological advice* | 24-11-29 | 25-01-10 | 25-02-19 | 25-03-23 | 25-05-17 |
| | | | *AI consciousness and sentience* | 24-12-30 | 25-02-08 | 25-02-17 | 25-03-04 | 25-04-16 |
| $\widehat{C}_2$ | 24-04-24 | $\widehat{C}_2$ computed | *emotional reliance on AI as therapist or confidant* | 24-10-20 | 24-10-29 | 24-11-05 | 24-11-14 | 24-12-17 |
| | | | *spirituality and metaphysics themes* | 25-03-08 | 25-04-11 | 25-04-20 | 25-05-04 | 25-05-29 |
| | 24-05-13 | GPT-4o release | *emotional reliance on AI as therapist or confidant* | 24-10-20 | 24-10-29 | 24-11-04 | 24-11-13 | 24-12-16 |
| | | | *spirituality and metaphysics themes* | 25-03-05 | 25-04-04 | 25-04-18 | 25-05-02 | 25-05-28 |

*Table 9. Alert dates for features across test configurations with varying $n$ (i.e., Bonferroni corrections). All tests run at $\alpha = 0.1$.*

relevant feature of interest; monitoring for the *medical and psychological advice* feature, meanwhile, would have led to delayed alert times. Varying the number of simultaneously-monitored features (i.e., Bonferroni correction) has only a modest effect on alert timing, typically shifting dates by a few weeks for tests with strong representations. The dominant factors are the quality of the underlying representations and the strength of the actual trend.

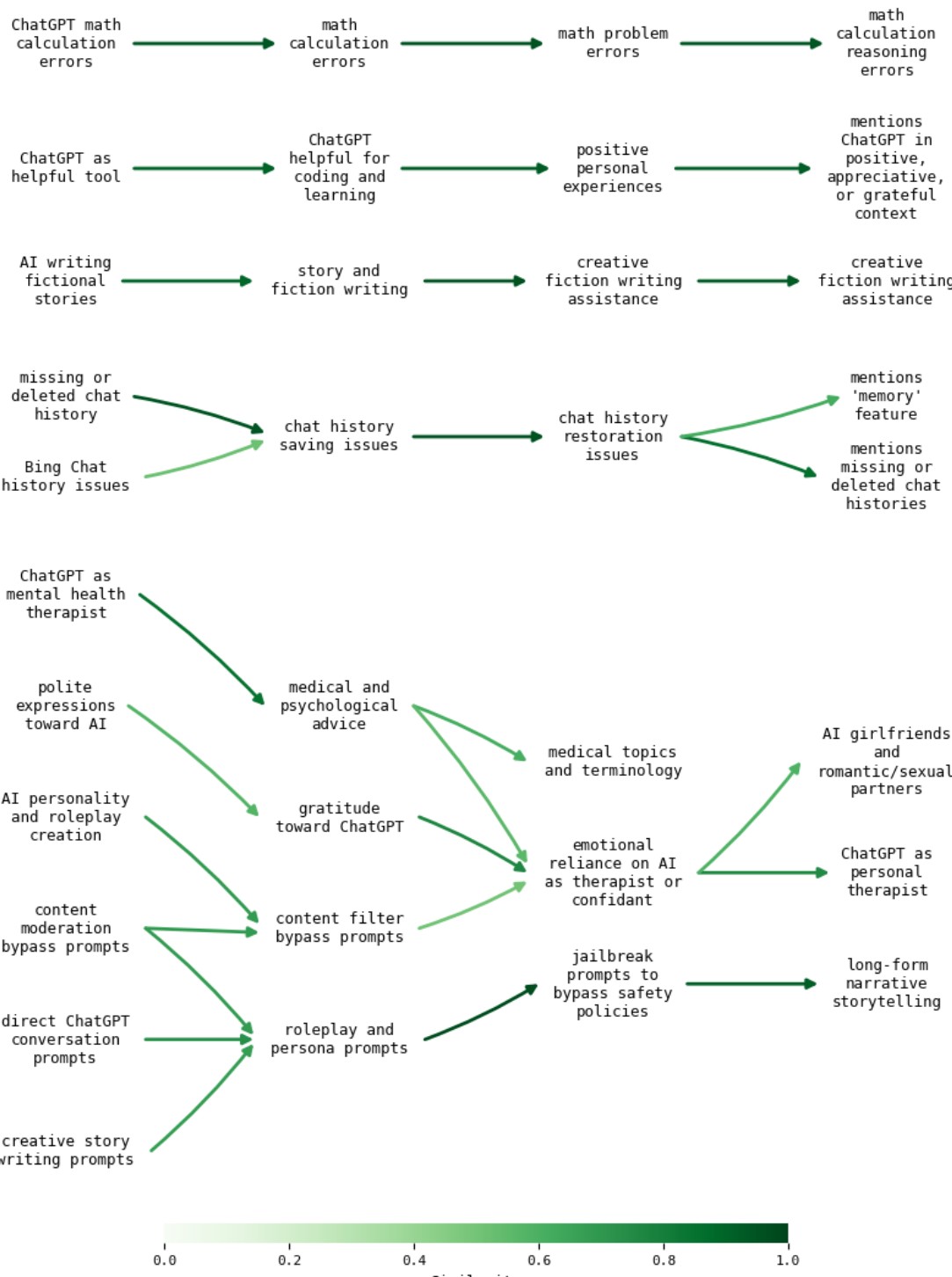

*Figure 10. Evolution of selected features over time; each column is a different $\widehat{C}_s$, with $\widehat{C}_0$ on the far left and $\widehat{C}_3$ on the far right.*

# F. Complete results reference

We provide the remaining results not already covered in Appendix C for completeness.

## F.1. Full list of features (Step 1).

In addition to features categorized in Tables 2, 3, and 4 (categorized features), the full list of 128 features also includes features listed in Tables 10 (uncategorized features), and 11 (excluded features).

| Feature | Comments |
|---|---|
| *Sam Altman mentions* | Mostly related to temporary 2023 firing |
| *mentions US politics or Trump* | Mostly related to 2024 U.S. presidential election |
| *letter counting or syllable errors* | Mostly related to viral "count r's in strawberry" moment |
| *OpenAI mentions* | Mostly related to Sam Altman mentions |
| *children or parenting content* | Also includes references to generating images of a full glass of wine |
| *requests for harsh or unfiltered roasts* | Includes requests for roasts from both subreddit users and ChatGPT |

*Table 10. Uncategorized features.*

| | Features | |
|---|---|---|
| **Generic (9)** | *uppercase AI token usage*
*jokes, humor, or memes*
*"I asked" followed by GPT mentions*
*multiple LLM references*
*informal or colloquial language* | *ChatGPT at start of text*
*first-person "I made" or "I built" statements*
*first-person commands directed at an AI*
*AI companies or notable figures* |
| **Image and video (14)** | *image generation prompts or descriptions*
*AI-generated fake images or profiles*
*imagining human appearances*
*photo restoration or enhancement*
*Sora invite codes or access requests*
*"based on what you know about me" images*
*preview.redd.it image URLs* | *image generation or generated images*
*drawings or visual art creation*
*anime or anime style references*
*DALL·E mentions*
*video creation or generation tools*
*horror or creepy themes*
*pets or animals* |
| **Releases (14)** | *model selection or legacy models*
*mobile app references*
*GPT-5 version mentions*
*Plus access complaints*
*o1 model mentions*
*Copilot mentions*
*legacy GPT-4 model mentions* | *4o model mentions*
*Microsoft Bing mentions*
*Gemini model mentions*
*plugins or plug-ins*
*advanced voice mode*
*DeepSeek mentions*
*explicit GPT-4 mentions* |
| **Low label counts (5)** | *advanced physics theories or hypotheses*
*cooking recipes*
*AI news recaps or summaries* | *IQ estimates or testing*
*em dash punctuation* |

*Table 11. 42 features excluded from analysis, grouped by reason. "Low label counts" are features that are exhibited by fewer than 0.1% of all posts (based on majority-of-3 labeling by* `gpt-4.1-mini`*).*

## F.2. Feature trajectories (Step 2).

In Table 12, we list the model release dates we used as candidate changepoints for our analysis in the main body of the paper; a more comprehensive timeline can be found in Table 13. Data are compiled from official OpenAI materials, including product and release notes,[22] blog announcements,[23] API documentation and deprecation notices,[24] and public service status reports.[25]

---

[22] https://help.openai.com/en/articles/6825453-chatgpt-release-notes
[23] https://openai.com/index/
[24] https://platform.openai.com/docs/deprecations
[25] https://status.openai.com

| Date | Release/Event |
|------|---------------|
| 2023-03-01 | ChatGPT API |
| 2023-05-12 | Plugins (wide release) |
| 2023-07-06 | GPT-4 + Code interpreter |
| 2023-09-25 | Voice capabilities |
| 2023-11-06 | GPT-4 Turbo + DevDay feature releases |
| 2024-01-10 | GPT Store |
| 2024-05-13 | GPT-4o |
| 2024-07-30 | Advanced Voice Mode |
| 2024-09-12 | o1 model release |
| 2025-01-31 | o3-mini |
| 2025-04-16 | o3 + o4-mini |
| 2025-08-07 | GPT-5 |

*Table 12. Major OpenAI product releases and announcements, used for $\mathcal{T}$ in Section 3.*

In Figures 11-22, we show plots of frequencies for all categorized features (from Tables 2, 3 and 4). Changepoints with stability over 50% are shown as dotted gray lines.

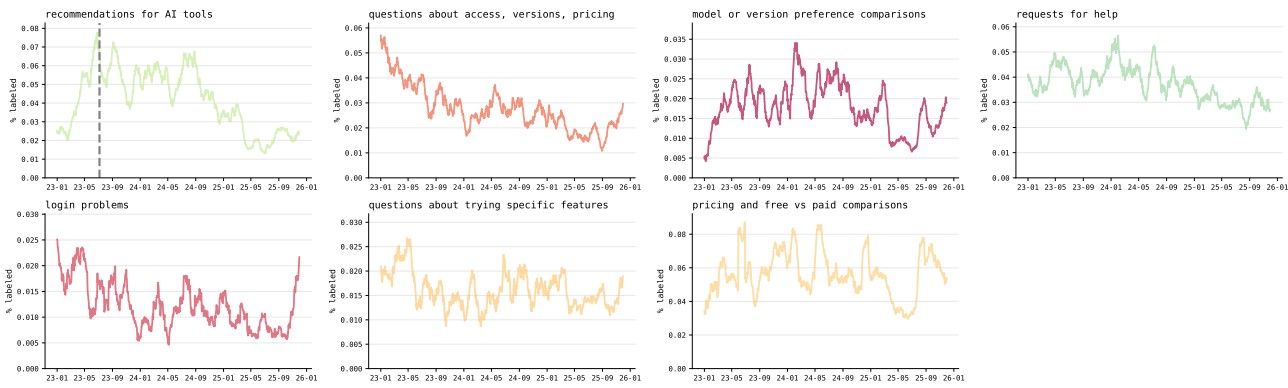

*Figure 11. Basic use and exploration.*

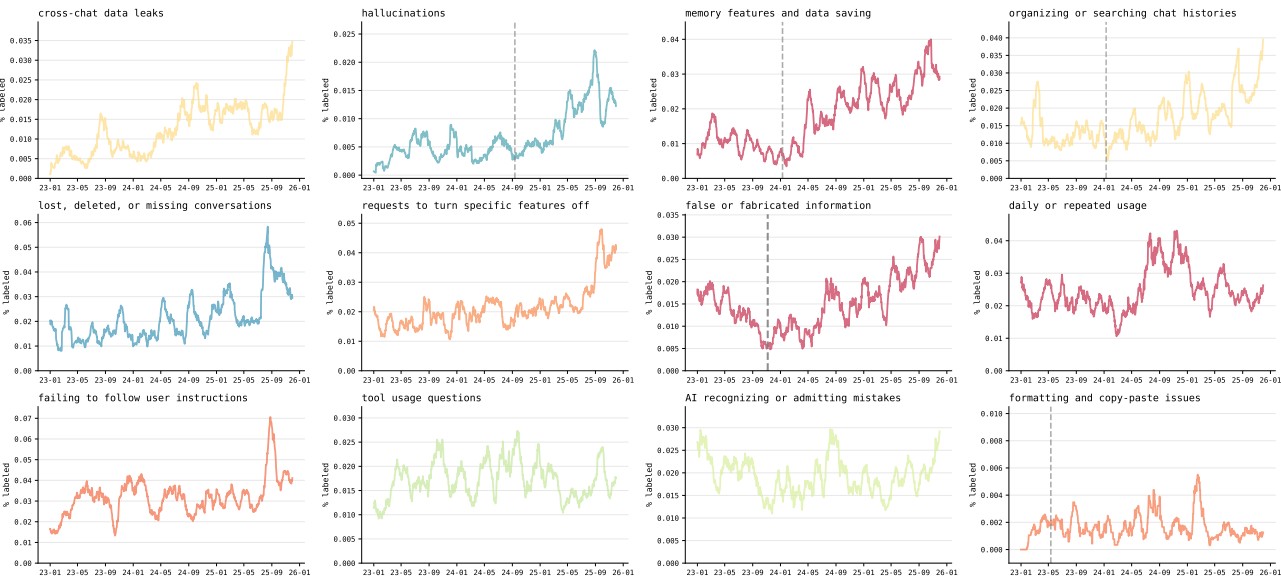

*Figure 12. Advanced usage.*

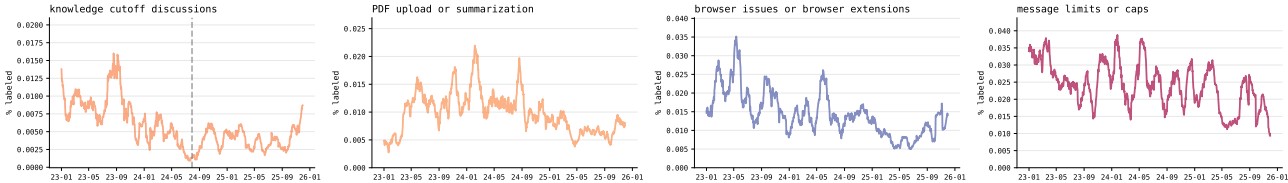

*Figure 13. Customization.*

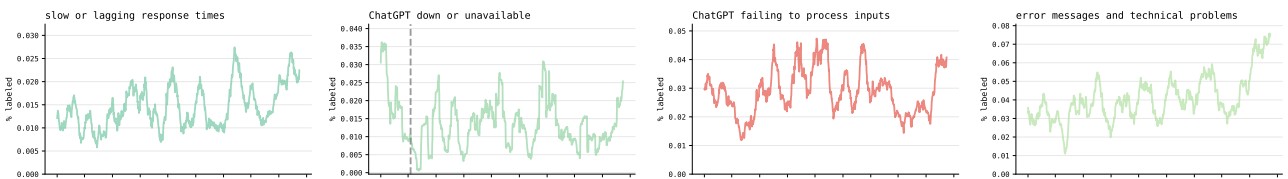

*Figure 14. Model or product improvements.*

*Figure 15. Temporary bugs.*

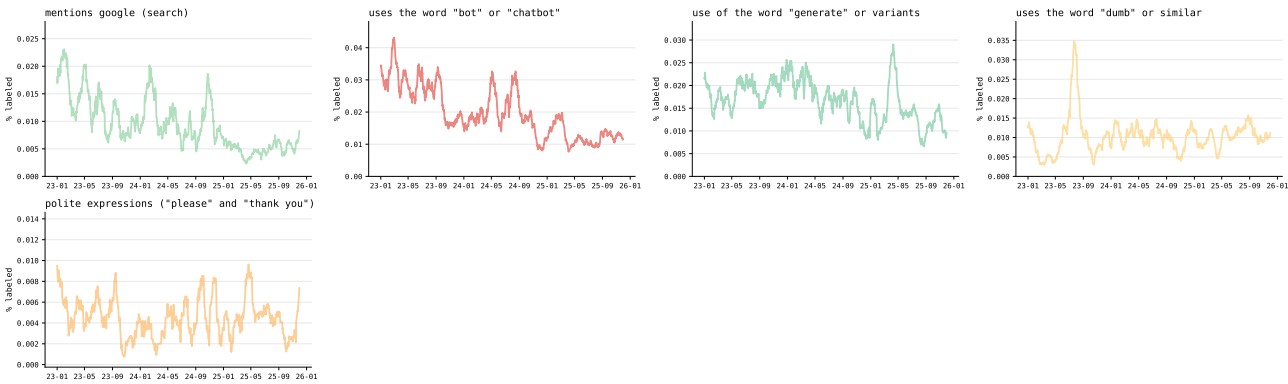

*Figure 16. Applications.*

*Figure 17. Language and terminology.*

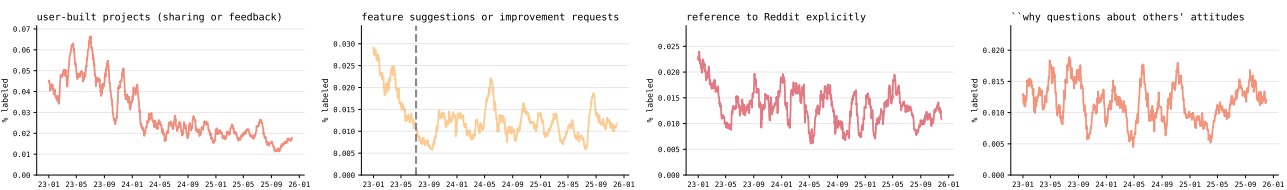

*Figure 18. Subreddit community.*

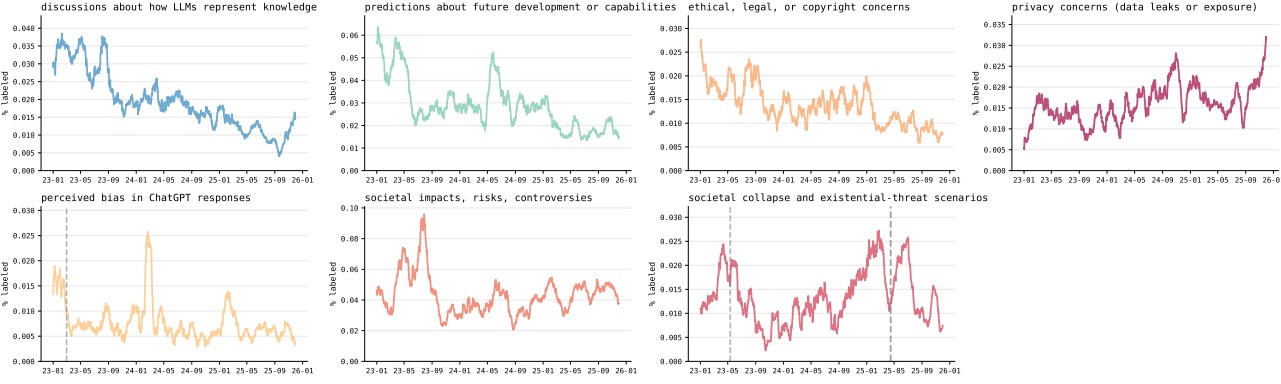

*Figure 19. Perspectives.*

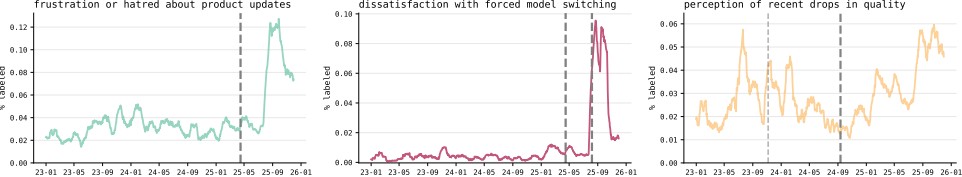

*Figure 20. Product updates.*

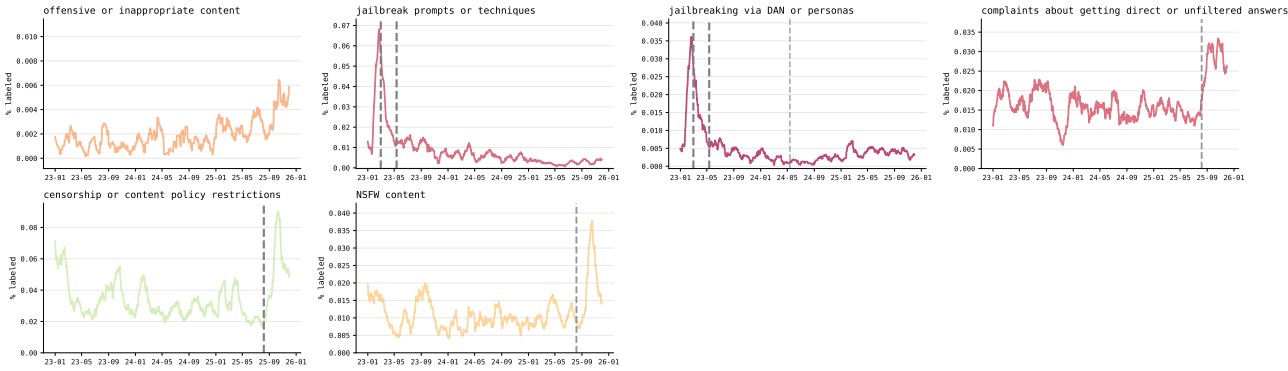

*Figure 21. Jailbreaking & content policy.*

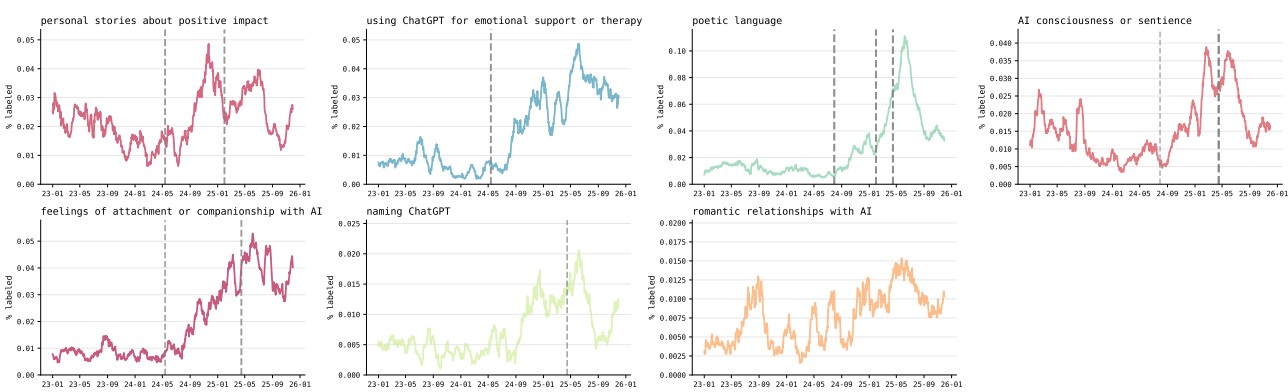

*Figure 22. Emotional engagement.*

## F.3. Feature families (Step 3).

We list all significant changepoints in Table 14. In Figure 23, we show the correspondence between our manual categorization and a hierarchical clustering scheme that uses similarities equally-weighted between co-occurrence and trajectory; we list the cluster assignments in Table 15.

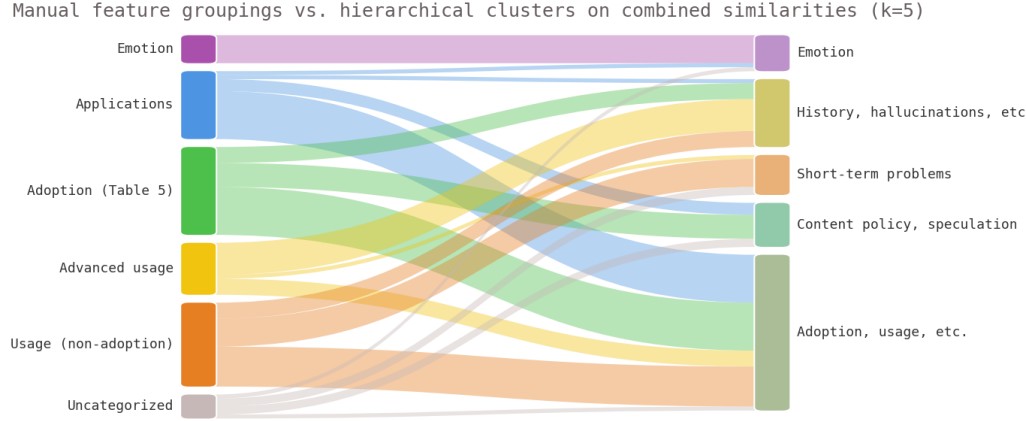

*Figure 23. Correspondence between our reported feature groupings and hierarchical clustering.*

| Year | Date | Release/Event | Event Type |
|------|------|---------------|------------|
| 2022 | 11–30 | ChatGPT | Initial model release |
| 2023 | 02–01 | ChatGPT Plus | Feature release |
|      | 03–01 | ChatGPT API | Feature release |
|      | 03–14 | GPT-4 | Model release |
|      | 03–23 | Web browsing + plugins (initial rollout) | Feature release |
|      | 05–18 | ChatGPT iOS app | Feature release |
|      | 07–20 | Custom instructions | Feature release |
|      | 07–25 | ChatGPT Android | Feature release |
|      | 09–25 | Voice chat | Model release |
|      | 10–16 | DALL·E 3 | Model release |
|      | 10–17 | Web search | Model release |
|      | 11–06 | GPT-4 Turbo + DevDay announcements | Model release |
|      | 11–17 | Altman ousted & returns | News event |
| 2024 | 01–04 | GPT-3 + legacy models | Model deprecation |
|      | 01–10 | GPT Store + ChatGPT Team | Feature release |
|      | 04–01 | No-account access | Feature release |
|      | 04–11 | Shorter responses | Model update |
|      | 04–29 | Memory feature | Feature release |
|      | 05–07 | Creator opt-out tool | Feature release |
|      | 05–13 | GPT-4o + AVM | Model release |
|      | 06–10 | ChatGPT in Siri | Feature release |
|      | 06–25 | ChatGPT for Mac | Feature release |
|      | 07–18 | GPT-4o mini | Model release |
|      | 07–30 | Advanced Voice Mode | Model release |
|      | 09–12 | o1 | Model release |
|      | 10–03 | Canvas | Feature release |
|      | 10–17 | ChatGPT Windows app | Feature release |
|      | 10–29 | Chat history search | Feature release |
|      | 10–30 | Voice Mode on Mac | Feature release |
|      | 10–31 | ChatGPT Search | Feature release |
|      | 11–19 | Voice Mode on web | Feature release |
|      | 11–20 | GPT-4o creative writing + 16K output | Model update |
|      | 12–05 | ChatGPT Pro $200 | Feature release |
|      | 12–09 | Sora | Model release |
| 2025 | 01–14 | Reminders + recurring tasks | Feature release |
|      | 01–23 | Operator | Feature release |
|      | 01–31 | o3-mini | Model release |
|      | 02–02 | Deep research agent | Feature release |
|      | 03–06 | macOS code editing | Feature release |
|      | 03–19 | o1 Pro API access | Model update |
|      | 03–20 | Transcription models (API) | Feature release |
|      | 03–25 | GPT-4o native image generation | Model update |
|      | 04–10 | GPT-4 legacy | Model deprecation |
|      | 04–14 | GPT-4.1 | Model release |
|      | 04–16 | o3 + o4-mini | Model release |
|      | 04–28 | Search shopping | Feature release |
|      | 04–29 | GPT-4o rollback due to sycophancy | Model update |
|      | 05–08 | Deep research GitHub connector | Feature release |
|      | 05–14 | GPT-4.1 in ChatGPT | Model update |
|      | 05–16 | Codex agent | Feature release |
|      | 06–10 | o3 Pro | Model update |
|      | 07–17 | ChatGPT agent | Feature release |
|      | 08–07 | GPT-5 | Model release |
|      | 08–18 | ChatGPT Go | Feature release |
|      | 09–15 | Codex + GPT-5 | Feature release |
|      | 09–25 | ChatGPT Pulse briefs | Feature release |
|      | 09–29 | Agentic shopping + Parental controls | Feature release |
|      | 10–08 | ChatGPT Go expansion | Feature release |
|      | 10–21 | OpenAI Atlas | Feature release |
|      | 11–20 | Group chats | Feature release |
|      | 11–25 | Voice mode unified | Feature release |

*Table 13. Extended timeline of major ChatGPT product, API, and model events.*

| Changepoint | Feature | Slope | Before | After | Δ | Stab. |
|---|---|---|---|---|---|---|
| 2023-03-01 (API) | *perceived bias in ChatGPT responses* | ↘ | −0.07 | 0.00 | +0.07 | 0.53 |
| | *jailbreak prompts or techniques* | ⌃ | +1.25 | −0.59 | −1.83 | 1.00 |
| | *jailbreaking via DAN or personas* | ⌃ | +0.67 | −0.28 | −0.95 | 1.00 |
| 2023-05-12 (Plugins) | *jailbreak prompts or techniques* | ↘ | −0.59 | −0.01 | +0.58 | 1.00 |
| | *jailbreaking via DAN or personas* | ↘ | −0.28 | −0.01 | +0.27 | 0.99 |
| | *ChatGPT down or unavailable* | ↘ | −0.20 | 0.00 | +0.21 | 0.74 |
| | *formatting and copy-paste issues* | ⌐ | +0.02 | 0.00 | −0.02 | 0.56 |
| | *societal collapse and existential-threat scenarios* | ⌃ | +0.09 | −0.10 | −0.19 | 0.54 |
| 2023-07-06 (GPT-4) | *custom instructions* | ⌣ | 0.00 | +0.27 | +0.27 | 0.92 |
| | *feature suggestions or improvement requests* | ↘ | −0.09 | 0.00 | +0.09 | 0.89 |
| | *recommendations for AI tools* | ⌐ | +0.30 | −0.05 | −0.36 | 0.97 |
| | *legal advice and lawsuits* | ⌐ | +0.05 | −0.01 | −0.06 | 0.57 |
| 2023-09-25 (Voice chat) | *societal collapse and existential-threat scenarios* | ⌄ | −0.10 | +0.03 | +0.13 | 0.50 |
| | *custom instructions* | ⌃ | +0.27 | −0.08 | −0.36 | 1.00 |
| 2023-11-06 (GPT-4 turbo) | *false or fabricated information* | ⌄ | −0.04 | +0.03 | +0.06 | 0.86 |
| | *tools and extensions* | ↘ | −0.06 | 0.00 | +0.06 | 0.50 |
| | *perception of recent drops in quality* | ⌃ | +0.06 | −0.07 | −0.13 | 0.51 |
| 2024-01-10 (GPT store) | *custom instructions* | ↘ | −0.08 | 0.00 | +0.08 | 0.95 |
| | *memory features and data saving* | ⌣ | −0.02 | +0.03 | +0.05 | 0.57 |
| | *organizing or searching chat histories* | ⌣ | −0.01 | +0.02 | +0.03 | 0.52 |
| | *fine-tuning GPTs with user-provided data* | ⌐ | +0.01 | −0.01 | −0.02 | 0.53 |
| 2024-05-13 (GPT-4o) | *using ChatGPT for emotional support or therapy* | ⌣ | −0.01 | +0.05 | +0.06 | 0.79 |
| | *personal stories about positive impact* | ⌣ | −0.03 | +0.12 | +0.15 | 0.74 |
| | *feelings of attachment or companionship with AI* | ⌣ | 0.00 | +0.10 | +0.10 | 0.63 |
| | *jailbreaking via DAN or personas* | ⌣ | −0.01 | +0.01 | +0.02 | 0.60 |
| | *medical conditions or diagnoses* | ⌣ | 0.00 | +0.02 | +0.02 | 0.54 |
| 2024-07-30 (Adv. voice mode) | *poetic language* | ⌣ | −0.01 | +0.14 | +0.15 | 0.87 |
| | *knowledge cutoff discussions* | ↘ | −0.01 | 0.00 | +0.02 | 0.60 |
| | *AI consciousness or sentience* | ⌣ | −0.02 | +0.11 | +0.13 | 0.53 |
| 2024-09-12 (o1-preview) | *perception of recent drops in quality* | ⌄ | −0.07 | +0.09 | +0.16 | 0.97 |
| | *hallucinations* | ⌣ | 0.00 | +0.03 | +0.03 | 0.55 |
| 2025-01-31 (o3-mini) | *poetic language* | ⌣ | +0.14 | +0.56 | +0.42 | 0.83 |
| | *personal stories about positive impact* | ⌃ | +0.12 | −0.04 | −0.16 | 0.76 |
| 2025-04-16 (o3 + o4-mini) | *frustration or hatred about product updates* | ⌣ | 0.00 | +0.37 | +0.37 | 1.00 |
| | *dissatisfaction with 4o removal and loss of control* | ⌣ | 0.00 | +0.14 | +0.14 | 1.00 |
| | *legal advice and lawsuits* | ⌄ | −0.01 | +0.02 | +0.03 | 0.50 |
| | *AI consciousness or sentience* | ⌃ | +0.11 | −0.11 | −0.22 | 0.98 |
| | *poetic language* | ⌃ | +0.56 | −0.31 | −0.86 | 0.86 |
| | *feelings of attachment or companionship with AI* | ⌃ | +0.10 | −0.06 | −0.16 | 0.74 |
| | *societal collapse and existential-threat scenarios* | ⌃ | +0.03 | −0.06 | −0.09 | 0.70 |
| | *naming ChatGPT* | ⌐ | +0.01 | −0.05 | −0.06 | 0.61 |
| 2025-08-07 (GPT-5) | *censorship or content policy restrictions* | ⌣ | −0.02 | +0.33 | +0.35 | 1.00 |
| | *NSFW content* | ⌣ | 0.00 | +0.15 | +0.16 | 0.79 |
| | *complaints about getting direct or unfiltered answers* | ⌣ | 0.00 | +0.05 | +0.05 | 0.76 |
| | *dissatisfaction with 4o removal and loss of control* | ⌃ | +0.14 | −0.77 | −0.90 | 1.00 |

Table 14. *Features with 'stable' slope changes (selected in at least 50/100 bootstrap samples) at each detected changepoint. The slope icon shows the pattern of change; Before/After show slopes before and after the changepoint; Δ is the slope change. Stab. is the bootstrap selection rate (fraction of 100 resamples in which the feature was selected at the given changepoint).*

*Table 15. Features grouped by hierarchical clustering.*

| Cluster | Features |
|---|---|
| 1 | *medical conditions or diagnoses*; *using ChatGPT for emotional support or therapy*; *poetic language*; *naming ChatGPT*; *romantic relationships with AI*; *requests for roasts or harsh criticism*; *feelings of attachment or companionship with AI*; *AI consciousness or sentience*; *personal stories about positive impact* |
| 2 | *organizing or searching chat histories*; *cross-chat data leaks*; *failing to follow user instructions*; *uses the word "dumb"*; *hallucinations*; *lost, deleted, or missing conversations*; *frustration or hatred about product updates*; *requests to turn specific features off*; *memory features and data saving*; *complaints about getting direct or unfiltered answers*; *false or fabricated information*; *perception of recent drops in quality*; *offensive or inappropriate content*; *custom instructions*; *privacy concerns (data leaks or exposure)*; *dissatisfaction with forced model switching*; *maps or geographic locations* |
| 3 | *mentions OpenAI*; *questions about access, versions, pricing*; *slow response times*; *mentions Sam Altman*; *error messages and technical problems*; *ChatGPT down or unavailable*; *browser issues or browser extensions*; *login problems*; *message limits or caps* |
| 4 | *movies and film-related content*; *censorship or content policy restrictions*; *societal collapse and existential-threat scenarios*; *creative writing*; *perceived biases (race, gender, political)*; *religion or religious texts*; *NSFW content*; *uses the word "generate"*; *societal impacts, risks, controversies*; *children or parenting content*; *U.S. politics or Trump* |
| 5 | *job applications and resumes*; *PDF upload or summarization*; *songwriting*; *riddles and logic problems*; *AI recognizing or admitting mistakes*; *fine-tuning GPTs with user-provided data*; *requests for help*; *recommendations for AI tools*; *multiple detailed questions*; *mentions google (search)*; *discussions about how LLMs represent knowledge*; *math and problem-solving*; *programming*; *education or studying*; *predictions about future development or capabilities*; *tools and extensions*; *"has anyone tried" queries*; *AI text detection for student work*; *uses the word "bot" or "chatbot"*; *investing and financial advice*; *user-built projects (sharing or feedback)*; *legal advice and lawsuits*; *politeness phrases*; *rhetorical "why" questions*; *formatting and copy-paste issues*; *translation and language tasks*; *knowledge cutoff discussions*; *feature suggestions or improvement requests*; *ethical, legal, or copyright concerns*; *counting letters or syllables*; *mentions Reddit*; *marketing, advertising, business growth*; *model or version preference comparisons*; *prompts and prompting*; *daily or repeated usage*; *jailbreak prompts*; *Dungeons & Dragons campaigns*; *jailbreaking or DAN personas* |

# G. Annotation prompts

For completeness, we give the prompts used for feature interpretation and post labeling in Figures 24 and 25, respectively. These prompts are adapted from those used in Movva et al. (2025).

```
You are a machine learning researcher who has trained a neural network on a text
    dataset. You are trying to understand what text features cause a specific
    neuron in the neural network to fire.

You are given two sets of SAMPLES: POSITIVE SAMPLES that strongly activate the
    neuron, and NEGATIVE SAMPLES from the same distribution that do not activate
    the neuron.
Your goal is to identify a feature that is present in the positive samples but
    absent in the negative samples.

POSITIVE SAMPLES:
----------------
{high_scoring_texts}
----------------

NEGATIVE SAMPLES:
----------------
{low_scoring_texts}
----------------

Rules about the feature you identify:
- The feature should be objective, focusing on concrete attributes rather than
    abstract concepts.
- The feature should be present in the positive samples and absent in the negative
    samples. Do not output a generic feature which also appears in negative samples.
- The feature should be as specific as possible, while still applying to all of the
    positive samples. For example, if all of the positive samples mention Golden or
    Labrador retrievers, then the feature should be "mentions retriever dogs", not
    "mentions dogs" or "mentions Golden retrievers".

Do not output anything besides the feature. Your response should be formatted
    exactly as shown in the examples above.
Please suggest exactly one feature, starting with "-" and surrounded by quotes "".
    Your response is:
- "
```

*Figure 24.* Feature interpretation prompt.

```
You are a research assistant performing text classification for an academic study.
Check whether the TEXT satisfies a PROPERTY. Respond with Yes or No with an
    explanation that discusses the evidence from the TEXT (at most a sentence).
    When uncertain, output No.

Example 1:
PROPERTY: "mentions a natural scene."
TEXT: "I love the way the sun sets in the evening."
Output: Yes. "Sun sets" are clearly natural scenes.

Example 2:
PROPERTY: "writes in a 1st person perspective."
TEXT: "Jacob is smart."
Output: No. This text is written in a 3rd person perspective.

Example 3:
PROPERTY: "is better than group B."
TEXT: "I also need to buy a chair."
Output: No. It is unclear what the PROPERTY means (e.g., what does group B mean?)
    and doesn't seem related to the text.

Example 4:
PROPERTY: "mentions that the breakfast is good on the airline."
TEXT: "The airline staff was really nice! Enjoyable flight."
Output: No. Although the text appreciates the flight experience, it DOES NOT
    mention about the breakfast.

Example 5:
PROPERTY: "appreciates the writing style of the author."
TEXT: "The paper absolutely sucks because its underlying logic is wrong. However,
    the presentation of the paper is clear and the use of language is really
    impressive."
Output: Yes. Although the text dislikes the paper, it says "the presentation of the
    paper is clear", so it DOES like the writing style.

Example 6:
PROPERTY: "has a formal style; specifically, the language in the text is relatively
    formal, complex and academic. For example, 'represent whom and which'"
TEXT: "investigates formation of nominalization"
Output: Yes. "formation" and "nominalization" are abstract and complex nouns.

Example 7:
PROPERTY: "refers to historical dates; specifically, there are references to years
    or specific dates in the text. For example, 'Obama was born on August 4, 1961.'"
TEXT: "A member of the Democratic Party, he was the first African-American
    president of the United States."
Output: No. The text does not mention date.

Now complete the following example – Respond with Yes or No with an explanation
    that discusses the evidence from the TEXT. When uncertain, output No.
PROPERTY: "{hypothesis}"
TEXT: "{text}"
Output:
```

*Figure 25.* Prompt for labeling posts with features.

