# OpenReview forum: "Three Years of r/ChatGPT: Societal Impact Evaluations from Social Media Data"
_ICML.cc/2026/Conference — ICML 2026 regular_

### Official Review · Reviewer_XY7j · 2026-03-07

**Soundness:** 4
**Presentation:** 3
**Significance:** 4
**Originality:** 3
**Overall Recommendation:** 5
**Confidence:** 4

**Summary:**

Summary:
This paper introduces the first longitudinal study of the Reddit forum r/ChatGPT, with over 137,000+ posts from December 2022 to November 2025. It examines the discourse of the general public discussing ChatGPT through the lens of social media and its impact on society, with two approaches:
- Retrospective approach: SAE-based featurization of posts into approximately 75 interpretable topics, monitored over time with a piecewise linear and Hawkes process model, with OpenAI releases as candidate changepoints
- Prospective approach: Real-time framework with anytime-valid sequential hypothesis tests, with statistical guarantees for FDR/FWER, even with the addition/removal of features

Key Contributions:
1) General-purpose framework for assessing the societal impact of consumer-level AI products through social media
2) Real-time monitoring framework with statistical guarantees for applications to dynamic and ever-changing data streams
3) New insights into the normalization and emotional enmeshing of ChatGPT through the years
4) Demonstration of the potential for prospective monitoring to inform the development of AI products, rather than waiting for potential issues to arise

**Compliance With Llm Reviewing Policy:**

Affirmed.

**Final Justification:**

This paper is a timely and well-motivated contribution that brings an important perspective on how to assess societal consequences of consumer AI tools through a longitudinal study design with formal guarantees and statistically significant results on normalization and engagement. Most of my concerns were successfully handled by the authors. Specifically, the authors managed to provide an insightful analysis of additional validation of their LLM annotation procedure (majority vote on multiple calls), justify reproducibility of feature selection with a reference to a specific methodology, confirm that the proposed method of prospective monitoring remains feasible under the present restrictions on using Reddit API, and, most importantly, provide new experiments comparing SAE representation of user posts with two other options (PCA and k-means) and demonstrating broadly similar features. Despite remaining concerns such as $\hat{C_0}$ failure for therapy detection and inherently unrepresentative nature of Reddit sample, the authors have addressed these limitations honestly without compromising the validity of the main findings reported in the paper. I recommend acceptance.

**Key Questions For Authors:**

The paper is clearly written and the methodology and experiments are sufficiently described. I do not have clarification questions for the authors.

**Limitations:**

Yes

**Strengths And Weaknesses:**

Soundness (strengths):
- Large, well-curated dataset: 137,154 posts from 89,346 unique users over a span of three years; steps taken in data cleaning are well-justified
- Formal basis for statistical methods: anytime-valid sequential hypothesis tests with guarantees of FDR/FWER control, with proofs included in appendix
- Candid admission of limitations
- Robustness of results over various configurations of monitoring settings
- Stability of representations: featurization needed to be updated only three times over a span of three years

Soundness (weaknesses):
- LLM-aided feature interpretation may lead to bias: Using GPT-4.1-mini for feature interpretation and sample annotation may lead to systematic errors or model-specific biases that are difficult to audit
- Initial featurization $\hat{C_0}$ fails for therapy detection regardless of $\alpha$ and $\beta$
-Manual feature curation may lead to subjectivity: removal of 53 of 128 features through a combination of automated and manual steps, including "manual inspection," is not fully reproducible and may affect results

Presentation (strengths):
- Narrative structure: engaging and easy to follow
- Figure 1 is engaging: the reader gets a sense of the importance of the paper’s contribution early on
- Positioning: effectively distinguishes the paper from topic modeling, event detection, longitudinal LLM evaluation, and crowdsourced evaluation research areas
- Algorithms 1-5 are well-written: build the reader’s intuition before the formal treatment in the appendix

Presentation (weaknesses):
- Overuse of appendix: important information is relegated to the appendix, making the 8 pages difficult to analyze without the appendix
- Discussion is too short: does not fully explore the implications for AI governance, platform regulation, and product design
- Figures in the appendices: the trajectory plots in the appendix (Figures 6-8) are too small and too dense – text is very small

Significance (strengths):
- Directly policy-relevant finding: showing the efficacy of such a public monitoring framework by illustrating that it could have monitored the increase in emotional engagement within October 2024, prior to OpenAI's public acknowledgment, is a strength of this framework's utility and relevance
- Timely and widely impactful problem: the ability to monitor societal impact for AI products, which are rapidly approaching the billion-user mark, is now more crucial for all stakeholders, including regulators, developers, and researchers
- The GPT-5 backlash: showing that more than half of the complaints following the release were related to emotional engagement contexts provides a concrete and actionable finding about the disconnect between OpenAI's expectations and user experiences
- Framework generalizability: the framework's methodology is centered around its ability to be product- and platform-agnostic, providing it with potentially wide-ranging utility for AI governance research

Significance (weaknesses):
- Practical deployment challenges: the utility of the monitoring framework for public monitoring is dependent on continued API access for both Reddit and social media platforms, which has become increasingly limited, and this factor is not discussed
- Potentially confirms existing suspicions: the findings may simply confirm existing suspicions for many AI product developers, and so the impact of this study may not be particularly impactful for changing existing practices

Originality (strengths):
- First longitudinal analysis of the r/ChatGPT community using three years of data and temporal variation
- New combination of existing techniques
- Application of SAE techniques for social science/policy
- Dynamic feature tracking with formal guarantees: allows for the addition/deletion of features from the set of monitored features at any time, depending on the observed data, without compromising valid Type I error control

Originality (weaknesses):
- All core techniques are existing techniques: SAE, anytime valid e-value testing, and topic modeling are all well-documented existing techniques
- Limited comparison of alternatives for featurization: the paper notes that other featurization techniques, such as other featurization algorithms, could be used instead of SAE, but it is not clear how much this affects the results, making it difficult to gauge the impact of the choice of using SAE specifically

---

> ### Author Rebuttal · Authors · 2026-03-31
>
> Thanks for the thorough review! We will continue to polish the presentation of the manuscript, and will utilize the additional allowed page in the camera-ready to add discussion. We discuss additional concerns below.
>
> **Soundness 1 – LLM feature interp.** LLM interpretation and annotation has limitations, but is increasingly accepted practice in computational social science settings, especially at the scale of our dataset — in fact, such an analysis likely would have been infeasible absent LLM annotation. See also, e.g., citations on L091-092.
>
> Additionally, while revising the manuscript after submission, we did try additional labeling schemes (majority-vote from three calls to the gpt-4.1-mini API, as well as one-shot labels and majority-votes from a handful of larger models). We manually inspected/annotated a small subset of posts to compare these labeling schemes; for our final labels, we chose majority-vote of three calls to gpt-4.1-mini, as they were comparable in quality to majority-vote of the larger models while being substantially cheaper. (As an additional sanity check, updating the labels from one-shot to majority-vote changed some numbers slightly, but did not affect the overall findings.)
>
> **Soundness 2 – Reproducibility/ manual feature curation.** We did make some explicit choices about what features to focus on (e.g., we choose not to analyze image generation-related features); however, we disagree that this is an issue regarding ‘reproducibility’ of our results. If our method from S2 was applied to a new dataset, for example, we would expect analysts to also exercise discretion about what features seem to be potentially relevant in the context of the features discovered for that dataset.
>
> Our quantitative results are reproducible in the sense that if a different analyst wanted to follow our method, they could (1) also generate their own features and compare them to the ones we found; (2) use our features and label posts with their own annotation scheme, and compare our findings with theirs; or (3) replicate the entire pipeline from scratch, and compare our findings with theirs.
>
> Since our main findings are focused on specific categories of features, we are confident that any of these approaches would find results that are consistent with ours. Different analysts may choose to focus on different subsets of posts, but the possibility of such a situation is mainly about whether additional findings might exist, rather than whether the ones we report here would change in light of that different choice.
>
> **Significance 1 — API access.** For ongoing monitoring, the current Reddit API still allows querying for ‘new’ posts within a subreddit, up to the ~900 most recent posts, so prospective monitoring moving forward is still feasible with standard APIs. That said, we share this concern, and will add some discussion on this in the paper.
>
> **Significance 2 — may confirm existing suspicions.** It is impossible to know what developers’ existing suspicions are, or were, at any point in time. That said, in Appendix B.2, we briefly cover some publicly-available information that suggests that sycophancy, emotional engagement, and related issues were not especially salient among researchers and developers, especially in the early phases of 4o’s release.
>
> **Originality 1 — existing techniques.** Yes, we agree that our primary contributions are not in methodological development! We do think that Propositions 4.1 and 4.2 are interesting consequences of recent theory, and were excited to adapt them to work for real-world application settings.
>
> **Originality 2 — featurization alternatives.** In initial exploration, we tried two classic methods as an alternative to SAEs for computing the featurizations. Specifically, we applied (1) PCA to post embeddings to find 64 principal components, treating the positive and negative directions of each principal component as a separate feature for 128 total, and (2) k-means clustering on post embeddings with k=128. The features identified by both of these methods were broadly similar. Using the approach described in E.4 for comparing SAE-PCA and SAE-k-means, no more than 10 (of 128) identified features in each featurization cannot be matched to at least one feature in the other.
>
> We chose to use SAEs for the final version of the paper for two reasons: first, because SAEs naturally allow each post to correspond to multiple features (unlike k-means), while also enforcing sparsity (unlike PCA); second, because based on manual inspection, we felt that SAE features were higher quality and more diverse than the others (this is consistent with quantitative findings from prior work – see, e.g., ablations in Movva et al.).

---

> > ### Author Rebuttal · Reviewer_XY7j · 2026-03-31
> >
> > My concerns have been adequately addressed.

---

### Official Review · Reviewer_yB5V · 2026-03-08

**Soundness:** 3
**Presentation:** 4
**Significance:** 4
**Originality:** 2
**Overall Recommendation:** 5
**Confidence:** 4

**Summary:**

This uses observational data (specifically, scraped posts from the widely used platform "Reddit" and associated metadata like scores) to understand recent historical trends in the adoption of LLM-based consumer AI technologies. The paper also proposes a future approach doing real-time monitoring of adoption trends and potentially concerning usage patterns (e.g., emotional dependency) going forward. The paper includes a number of careful quantitative analyses and robustness checks, and also contributes by more generally documenting key events in the timeline of 2023-2026 AI roll out.

**Compliance With Llm Reviewing Policy:**

Affirmed.

**Final Justification:**

The rebuttal straightforwardly addressed my major questions. It seems realistic that the authors can incorporate these details into final copy.

**Key Questions For Authors:**

- Why were Pushshift and API data mixed?
- How might this kind of project proceed if APIs are increasingly gated?
- Were any other simple baseline analysis considered?

**Limitations:**

Yes.

**Strengths And Weaknesses:**

## Soundness

In terms of technical soundness, the paper includes and explains a number of seemingly appropriate quantitative methods. The goal here is mainly descriptive insight and a framework for future analyses.

Data sampling choices seem mostly good appropriate. The paper should probably explain a bit more how/why Pushshift data and API data were mixed, how this work (and especially future monitoring) might be affected by API access, policy issues around Pushshift-style data, etc (more on this below in "Questions"). Exclusion and filtering choices are reasonable and the actual dataset seems appropriate for answering the RQs at hand. Data-related limitations are discussed directly in the paper and the work does not try to overclaim.

In terms of technical choices in the analysis, it might be useful to comment on how the use of SAE-based featurization compares to older approaches in computational social science for feauturizing this kind of data (simpler word counts, hand-crafted language features, etc.). It would also be helpful to comment on the viability of using manual or semi-automated labeling for these RQs.

Overall, soundness is a strength.

## Presentation

The design of figures and tables was generally very clear. I expect readers will not have major issues following the analysis narrative or interpreting specific results. Overall, a strong aspect of the paper with no major concerning or confusing elements.

## Significance

The draft does a good job convincing readers of the significance of the issues at hand (understanding consumer AI adoption using a variety of methods; going beyond reports provided by labs). Personally, I do not think this paper has major weaknesses with respect to significance. I could see some readers expecting more contribution around the implementation of a "live analysis feature" (I did not find Algorithm 1 particularly helpful compared to just describing this process in prose, and I found the live update results less convincing than the descriptive insights -- but still useful!).

## Originality

This specific analysis is novel. As the paper notes, a number of studies use Reddit (or similar data) to answer these kinds of questions and there is a broader literature of computational social science. While I don't think the paper is missing a key piece of RW with overlapping or related insights, some readers might be surprised to see minimal engagement with related works in venues like CHI or ICWSM (for instance, many of the works mentioned in the Baumgartner et al. cite). This could likely be addressed with a bit more discussion of the context of that paper and the history of pushshift more generally.

---

> ### Author Rebuttal · Authors · 2026-03-31
>
> Thanks for your review and the feedback! We will work on the presentation of Alg 1 and S4 in general, and add some of the suggested references.
>
> Regarding questions:
>
> **API/Pushshift.** The Reddit API will return full post data, given a post’s unique ID; however, it does not allow querying for (e.g.) all posts, or all post IDs, within a certain time window. We use Pushshift to get the post IDs of interest, then enrich it with the API to ensure that comments and up/downvotes are collected correctly.
>
> Specifically, Pushshift data is stored as a snapshot of post data at the time that Pushshift’s indexer collects the data. This means that, e.g., while comments and upvotes/downvotes often accumulate over the course of several days, Pushshift data only includes the comments and upvotes/downvotes that were present at the time of indexing. Since our analysis makes use of these counts, we use the API to ensure that they were consistent with their actual state on the platform.
>
> **Gated APIs.** This is also a concern we have. For platforms less adversarial than, e.g., Twitter, we would like to remain hopeful that engaging with official channels (e.g., Reddit4Researchers) is an option in the long-run. (The current Reddit API still allows querying for ‘new’ posts within a subreddit, up to the ~900 most recent posts, so prospective monitoring moving forward is still feasible with standard APIs.)
>
> More ambitiously, one might imagine a completely separate platform that is purpose-built for collecting this type of user feedback — for instance, a government or nonprofit agency that runs something like VAERS for AI systems — though that raises other concerns, e.g., whether users would leave feedback at all.
>
> **Other baseline analysis.** We did not explicitly try the more classical computational social science methods mentioned (word counts or hand-crafted features), although we did try two more classic methods as an alternative to SAEs for computing the featurizations. Specifically, we applied (1) PCA to post embeddings to find 64 principal components, treating the positive and negative directions of each principal component as a separate feature for 128 total, and (2) k-means clustering on post embeddings with k=128. The features identified by both of these methods were broadly similar. Using the approach described in E.4 for comparing SAE-PCA and SAE-k-means, no more than 10 (of 128) identified features in each featurization cannot be matched to at least one feature in the other.
>
> We chose to use SAEs for the final version of the paper for two reasons: first, because SAEs naturally allow each post to correspond to multiple features (unlike k-means), while also enforcing sparsity (unlike PCA); second, because based on manual inspection, we felt that SAE features were higher quality and more diverse than the others (this is consistent with quantitative findings from prior work – see, e.g., ablations in Movva et al.).
>
> Regarding labeling approaches — we felt that manual labeling would be infeasible given the volume of data (137k posts, with 4 candidate features per post). While revising the manuscript after submission, we did try additional labeling schemes (majority-vote from three calls to the gpt-4.1-mini API, as well as one-shot labels and majority-votes from a handful of larger models). We manually inspected/annotated a small subset of posts to compare these labeling schemes; for our final labels, we chose majority-vote of three calls to gpt-4.1-mini, as they were comparable in quality to majority-vote of the larger models while being substantially cheaper. (As an additional sanity check, updating the labels from one-shot to majority-vote changed some numbers slightly, but did not affect the overall findings.)

---

> > ### Author Rebuttal · Reviewer_yB5V · 2026-03-31
> >
> > Thanks for the these responses. The authors have pretty much answered all the questions I had previously and these answers help to strength the contribution (especially clarify minor design decisions that might affect results).

---

### Official Review · Reviewer_XKu4 · 2026-03-12

**Soundness:** 3
**Presentation:** 3
**Significance:** 3
**Originality:** 2
**Overall Recommendation:** 4
**Confidence:** 3

**Summary:**

The paper studies how public discussion about ChatGPT changed over three years by analyzing posts from the r/ChatGPT subreddit. Instead of measuring actual product usage, the authors use Reddit posts as a proxy to track major themes over time, such as everyday utility, technical help, emotional support, and attachment. They find that ChatGPT increasingly appears to have become a normalized daily tool, while posts related to emotional reliance and companionship grew notably, especially around the period after GPT-4o. The paper’s main contribution is both this empirical observation and a monitoring framework meant to detect such emerging societal trends early.

**Compliance With Llm Reviewing Policy:**

Affirmed.

**Final Justification:**

After the rebuttal, I remain positive about acceptance. The authors have responded in detail to all my comments, and while some of these topics remain debatable, the paper has enough merit to warrant acceptance in my opinion.

**Key Questions For Authors:**

See the weaknesses above

**Limitations:**

Yes

**Strengths And Weaknesses:**

Strengths:

Soundness
1. The paper analyzes a large, multi-year dataset, which makes its temporal observations more credible.
2. The monitoring setup is methodologically grounded rather than purely descriptive, which strengthens the technical contribution.
Presentation
1. The paper has a clear structure from data collection, to retrospective analysis, to real-time monitoring.
Originality
1. The paper studies societal impact through longitudinal user discourse rather than standard benchmark evaluation.
2. The combination of topic discovery with an early-warning monitoring framework is a novel angle as far as I know.
Significance
The paper addresses an important question about how widely deployed AI systems affect users over time.


Weaknesses:

Soundness
1. As acknowledged by the authors, the paper uses Reddit posts as a proxy for real-world ChatGPT use, which limits how strongly the findings can be generalized.
2. Its strongest claims rely on time alignment with product releases, but that is not enough to establish causality.
3. The emotional findings are not very convincing, as it may be an accidental finding or cherry picked, a wider analysis should be conducted.

Presentation
1. The framing sometimes sounds broader than the evidence; the paper studies subreddit discourse, not societal impact in a full population-level sense.
Originality
1. While the application domain is timely, parts of the technical pipeline build on fairly standard components for topic discovery and temporal analysis.
2. The novelty is more in the combination and application than in a fundamentally new method.
Significance
1. Because the analysis is based on a single online community, it is unclear how far the main conclusions extend beyond that setting.
2. The emotional-support finding is important, but the paper does not fully show how actionable its monitoring framework would be in real deployment.

---

> ### Author Rebuttal · Authors · 2026-03-31
>
> Thanks for the careful review!
>
> **Soundness 1: Reddit as proxy.** Yes, as we discuss, this is unfortunately a fundamental limitation of any evaluation of real-world usage conducted outside of OpenAI. We aren’t trying to claim that the distribution of topics on r/ChatGPT necessarily matches the distribution of actual usage, just that subreddit posts are an important signal as-is.
>
> **Soundness 2: Not enough to establish causality.** It’s true that we are not aiming to establish capital-c Causality in the formal, causal inference sense; as we discuss in Footnote 4, doing so is consistent with our framework, but we felt that conducting such an analysis would have gone beyond the scope of this paper (especially given the limitations of the data, as you mention above, which would necessarily cap the strength of any causal claim that we could make). We think that the observational/correlational statement of our findings is still sufficiently interesting even without establishing formal ‘causality.’
>
> That said, in lieu of the ITS procedure, we have added a lightweight test for identified changepoints as follows. We first randomly split all data and use half to fit changepoints (Eq 1); then, using the other half, we use a Wald test (with Newey-West HAC errors) to test for structural breaks at each identified changepoint for every feature. After Bonferroni correcting over both the number of features and the number of identified changepoints, the p-values for the existence of a structural break at the GPT-4o launch for the therapy and attachment/companionship features are on the order of 1e-7 and 1e-11, respectively. We’ll add these details to the updated manuscript.
>
> **Soundness 3: Emotional findings….  may be cherry picked.** The strength of our findings regarding emotional engagement were surprising to us. However, we are confident in our claims about these features for two reasons. First, in the retrospective analysis, we analyze transcripts based on which posts were labelled as exhibiting specific features (rather than which posts ‘activated’ for those features, which may have been artifacts of the unsupervised learning process). It would be highly unlikely for labelling errors to accumulate in a way that would ‘accidentally’ lead to our observed transcripts.  Second, features corresponding to therapy and attachment/companionship consistently appear across different ways that featurizations are computed. This includes other unsupervised methods (e.g., PCA and k-means clustering), as well as across different slices of time (e.g., in the featurizations ‘recomputed’ over time from S4). We are happy to discuss further if there are other kinds of wider analyses that you might find convincing.
>
> **Presentation 1: Framing.** We appreciate this feedback — we don’t mean for our results to represent a comprehensive or exhaustive accounting of all societal impacts, but rather to make the case that social media data should be thought of as a source to measure some societal impacts (that might otherwise not be known). We’re stating this more clearly in our revisions.
>
> **Originality:** Yes, we agree that our primary contributions are not in methodological development! We do think that Propositions 4.1 and 4.2 are interesting consequences of recent theory, and were excited to adapt them to work for real-world application settings.
>
> **Significance 1: Single online community.** We agree that it is unclear the degree to which our substantive findings from s3 are applicable to other products beyond ChatGPT; analyzing other online communities may yield different findings, and in fact we expect that other products likely do have different types of impacts. What we do think extends beyond ChatGPT specifically is the general principle of using social media data as a lens for understanding the impact of AI products (and new technology more broadly).
>
> **Significance 2: Actionability.** It’s impossible to know, counterfactually, what types of actions might have been taken in a world where we had been running our monitoring algorithm and raised an alert in October. For instance, perhaps OpenAI may have adjusted decisions around rollbacks, model updates, or content policy; at the same time, perhaps public knowledge could have increased pressure for OpenAI to tread more carefully around emotional engagement, or for other actors (e.g. policymakers) to pay closer attention (see also response to 8LMo w3).

---

> > ### Author Rebuttal · Reviewer_XKu4 · 2026-04-02
> >
> > Thanks for your detailed rebuttal. I remain positive about this work.

---

### Official Review · Reviewer_8LMo · 2026-03-13

**Soundness:** 2
**Presentation:** 3
**Significance:** 3
**Originality:** 3
**Overall Recommendation:** 4
**Confidence:** 4

**Summary:**

In this paper, the authors conducted a longitudinal study of the r/ChatGPT subreddit to investigate the discourse around ChatGPT by everyday users. To support this work, the paper introduced both an ex post facto and real-time framework for using social media to study the sociotechnical impact of the adoption of AI tools among everyday users. Using major model and product releases as temporal anchors, the analysis surfaces several meaningful patterns, most notably that discourse around emotional attachment to ChatGPT rose sharply and immediately following the GPT-4o release.

**Compliance With Llm Reviewing Policy:**

Affirmed.

**Key Questions For Authors:**

1. How did the paper determine that members of the subreddit are non-expert users?

**Limitations:**

Some limitations are addressed in the paper.

**Strengths And Weaknesses:**

## Strengths

The paper does several things well, including that:

1. It is timely and addresses an important set of questions.
2. The community would also find the prospective component of the
framework useful in studying the societal impact of AI adoption.

## Weaknesses

Despite these contributions, there are several fundamental issues
this paper needs to address before it is ready for publication.

### 1. Lack of qualitative engagement with the data

The epistemic distancing from the people whose experiences this paper
claims to illuminate is concerning. Consider that some of those posts
almost certainly contain people describing self-harm ideation they
shared with ChatGPT because they had no one else to tell. People
describing how an AI became their primary emotional relationship.
This paper's methodology renders all of that invisible except as a
statistical signal. Given this context and considering that the core
contribution of this work is not a novel theoretical machine learning
architecture, this lack of qualitative engagement with said user data
needs to be remediated. While the paper notes IRB exemption, basic
research judgment requires engagement with humans as participants
not objects, even in the course of aggregate extraction and analysis
of intimate human experience.

### 2. Motivation of the study

Considering that this is a very sensitive topic, the repeated
emphasis on being "the first longitudinal study of r/ChatGPT" is
peculiarly intellectually hollow, when the subject matter involves
people's mental health struggles, emotional vulnerabilities, and
intimate relationships with technology. This framing comes across
as unnecessary and highlights a mismatch of motivation. The right
framing should involve empathically engaging with the issue at hand
and foregrounding the human stakes of the findings.

### 3. What responsibilities are expected of us as researchers given the findings from this research?

Given that this paper itself surfaces evidence of people in emotional
distress, what concrete responsibilities follow for researchers,
platform designers, and AI companies? This would make this work more
actionable, even more so now that everyday users are concerned about
the societal impacts of AI across multiple facets, including job
loss, advanced manipulation, and deskilling. For instance, the paper
detected a rising signal of emotional distress months before OpenAI
acknowledged it publicly. If researchers can detect this, what are
they obligated to do with that knowledge? It would be worth
addressing this in some form in the paper.

### 4. How did the paper determine that members of the subreddit are non-expert users?

Considering the limited reported qualitative engagement with the
data, how did the paper determine that the subreddit comprised
non-expert users? There is a rich body of scholarly work on social
media analysis across multiple venues that could help with providing an
empirical backing to this claim, if at all it is necessary to have
this claim in the first place. This framing ties back to the theme
of epistemic distancing and of expert researchers studying unknowing
non-expert users.

## Overall Assessment

The main concern with this paper is about lack of meaningful qualitative engagement with user data.

---

> ### Author Rebuttal · Authors · 2026-03-31
>
> Thanks for the thoughtful review! We really appreciate the feedback on these aspects of our work, as the crux of our motivation in this paper is precisely to broaden which voices are considered useful in AI evaluation, and which types of impacts are considered worthy of attention — we think there are ‘human stakes’ that are essentially ignored by current evaluation practice. You’re right that our current manuscript can do a better job at expressing these stakes, and are making edits to that end.
>
> We discuss each of the concerns you raise further below.
>
> **On qualitative engagement (w1) -**
>
> We are in agreement that there is something inevitably lost when going from individual experiences to aggregate statistics. From our point of view, aggregate statistics provide value, but they are not the entire story, and we do not mean for them to substitute for the myriad personal narratives that we draw from. While quantitative analysis is necessary given the volume of data we work with, the process of working on this project did also involve a lot of close-reading of individual posts — which were sometimes heartwarming, sometimes distressing, and everything in between. We chose not to discuss examples of specific posts in the manuscript in order to respect anonymity, and to avoid broadcasting individual users’ posts beyond the audience they were intended for (i.e., the subreddit).  For the same reasons, we also did not conduct further user-level analysis (e.g., collecting information about what other subreddits that particular users were posting in, which is common in other studies using Reddit data to study mental health).
>
> That said, we would love to learn what qualitative engagement with user data looks like to you. We’ve added the reasoning above to the paper, and are very open to considering further edits; please let us know if you have ideas!
>
> **On motivation (w2) -**
>
> We think it’s important to retain the framing as a longitudinal study of r/ChatGPT, because that accurately describes the study’s focus, novelty, and initial motivation. That said, we agree it’s important to foreground human impacts more than the submitted manuscript currently does, and are working to make revisions to this effect.
>
> We also hope that future researchers will continue to study questions regarding mental health using r/ChatGPT data. We don’t intend our findings to be the authoritative, final word on the topic of emotional engagement with ChatGPT on this subreddit, especially as we are not experts in mental health; to the contrary, we think of our findings as a starting point for further study that can paint a fuller picture of these impacts.
>
> **On responsibilities (w3) -**
>
> We’re glad you raised this point! While all of our analysis was done post-hoc (in late 2025), our results suggest that, *if anyone had been paying attention*, we would have had a better sense of (e.g.) changes in patterns of emotional engagement as they were developing in mid-late 2024. While it’s impossible to know what OpenAI did and didn’t know at different points in time (see brief discussion in Appendix B.2), we think that this information suggests that OpenAI could or should have made different decisions (e.g., about rollbacks, content policy, or model updates), which seem clearly suboptimal in hindsight.
>
> One key responsibility we see as members of the research community, therefore, is exactly to *pay attention*. We would like to work towards a world where, as a society, we have a better collective understanding of how AI impacts are unfolding, which can help us, e.g., make better policy or more effectively hold corporations accountable. More precisely, our job as researchers is to develop tools to measure the impacts of these products, and to communicate them clearly; that’s exactly what this paper is seeking to do. We’ll add additional discussion that states this more explicitly in a future revision.
>
> **On ‘non-expert’ users (q1/w4) -**
>
> By ‘non-expert’ users, we mean users who are not AI researchers or developers, and thus have little-to-no input on standard AI evaluation pipelines (of course, users are ‘experts’ in their own experiences, which is why we wanted to study this data to begin with). We think it is very reasonable to say that r/ChatGPT users are ‘non-experts’ in this sense — though if you have references in mind, we’re happy to consult them as well. We’ll clarify this more explicitly in revisions.

---

> > ### Author Rebuttal · Reviewer_8LMo · 2026-04-04
> >
> > Thank you for the rebuttal. The response to W3 is appreciated.
> >
> > **On W1 (qualitative engagement)**, see references below for methodological guidance.
> >
> > The most important point is that this paper already has posts labeled as therapy and companionship from the SAE analysis. A concrete actionable step would be to select ten to fifteen posts from each cluster posted in the weeks immediately following the GPT-4o release, read them closely, and add two to three paragraphs to the paper paraphrasing what people were actually writing during that period. Were users distressed, curious, dependent, relieved? What kinds of situations were they describing? This would meaningfully deepen the human dimension of the paper without additional burden on the authors.
> >
> > **On W4 (non-expert users)**, this is a categorization claim without sufficient empirical proof. It also contributes little to the conceptual foundation or findings of the paper. Simply using "users" throughout would be a more precise and accurate description.
> >
> > **References**
> >
> > Ammari, T., Schoenebeck, S., & Romero, D. (2019). Self-declared throwaway accounts on Reddit: How platform affordances and shared norms enable parenting disclosure and support. *Proceedings of the ACM on Human-Computer Interaction, 3*(CSCW), 1–30. https://doi.org/10.1145/3359237
> >
> > Proferes, N., Jones, N., Gilbert, S., Fiesler, C., & Zimmer, M. (2021). Studying Reddit: A systematic overview of disciplines, approaches, methods, and ethics. *Social Media + Society, 7*(2). https://journals.sagepub.com/doi/10.1177/20563051211019004
> >
> > Harvey, D., Rayson, P., Lobban, F., Palmier-Claus, J., Dolman, C., & Jones, S. (2025). Navigating hypersexuality in bipolar: Insights from a corpus-assisted discourse analysis of Reddit posts. *Sage Open Medicine*. https://journals.sagepub.com/doi/10.1177/09636625221146453

---

> > > ### Author Response · Authors · 2026-04-07
> > >
> > > Thanks for the response and references!
> > >
> > > **W1-** We're happy to implement your recommendation and agree that it will strengthen the paper. (In case you were curious, a very high-level preview is that most therapy/companionship posts after the 4o release are generally positive and tend to describe examples of how ChatGPT was helpful - e.g., optimistic/impressed/grateful. Immediately following the GPT-5 release, on the other hand, there is a lot of betrayal/grief/anger.)
> > >
> > > **W4-** We'll edit the abstract to remove the reference to "non-experts" and rework that prose to be more specific. We do think that "non-AI researcher/developer" is conceptually important in the context of AI evaluation -- i.e., in contrast to who typically designs benchmarks and sets evaluation priorities -- and want to emphasize that contrast, but it's useful feedback that "experts" is loaded. We'll also put "experts" in scare quotes on L704.
> > >
> > > Thanks again for engaging on this, we really do appreciate it!

---

### Decision · Program_Chairs · 2026-04-30

**Decision:**

Accept (regular)

**Comment:**

This paper presents a longitudinal study of the r/ChatGPT subreddit to understand discourse around ChatGPT among everyday users and how they are impacted by it. The work introduces both a retrospective analysis and a prospective, real-time monitoring framework, and identifies several notable trends, including increased normalization of ChatGPT as a daily tool and a rise in emotional attachment following the GPT-4o model release.

The paper is widely viewed as timely, well-motivated, and focused on an important real-world problem. Reviewers highlight the value of the large-scale, multi-year dataset and the combination of topic discovery with a monitoring framework that could enable early detection of emerging societal impacts. The proposed framework is useful for studying AI adoption and informing governance, and the empirical findings --particularly around emotional dependence -- are informative and potentially impactful. The presentation is generally clear, and the methodology is conceptually rigorous and technically sound.

Reviewers raised concerns regarding the lack of qualitative engagement with user data, the use of Reddit as a proxy for real-world usage, the lack of causal identification in time-aligned analyses, the framing of the motivation (which could better foreground human stakes rather than emphasize novelty), as well as questions about the generalizability of findings beyond a single online community. These concerns were addressed in the authors’ rebuttal. The authors constructively engaged with the feedback, clarified methodological choices, provided additional statistical validation for key findings, and committed to incorporating qualitative analysis and improving the framing around human impacts. These responses were acknowledged positively by the reviewers, with several indicating that their concerns were fully or largely resolved.

Overall, this is a strong and timely paper with clear merit and useful insights into a real-world issue.